# Fast Encoder-Based 3D from Casual Videos via Point Track Processing

**Yoni Kasten**[1]  
[1]NVIDIA Research

**Wuyue Lu**[2]  
[2]Simon Fraser University

**Haggai Maron**[1,3]  
[3]Technion

## Abstract

This paper addresses the long-standing challenge of reconstructing 3D structures from videos with dynamic content. Current approaches to this problem were not designed to operate on casual videos recorded by standard cameras or require a long optimization time. Aiming to significantly improve the efficiency of previous approaches, we present TRACKSTo4D, a learning-based approach that enables inferring 3D structure and camera positions from dynamic content originating from casual videos using a single efficient feed-forward pass. To achieve this, we propose operating directly over 2D point tracks as input and designing an architecture tailored for processing 2D point tracks. Our proposed architecture is designed with two key principles in mind: (1) it takes into account the inherent symmetries present in the input point tracks data, and (2) it assumes that the movement patterns can be effectively represented using a low-rank approximation.TRACKSTo4D is trained in an unsupervised way on a dataset of casual videos utilizing only the 2D point tracks extracted from the videos, without any 3D supervision. Our experiments show that TRACKSTo4D can reconstruct a temporal point cloud and camera positions of the underlying video with accuracy comparable to state-of-the-art methods, while drastically reducing runtime by up to 95%. We further show that TRACKSTo4D generalizes well to unseen videos of unseen semantic categories at inference time. Project page: *https://tracks-to-4d.github.io*.

## 1   Introduction

Predicting 3D geometry in dynamic scenes is a challenging problem. In this problem setup, we are given access to multiple images of a scene taken sequentially, e.g., from a monocular video camera, where *both* the content in the scene and the camera are moving. Our task is to reconstruct the dynamic 3D positions of the points seen in the images and the camera poses. This fundamental problem has gained significant interest from the research community over the years [7, 37, 26, 64], mainly due to its important applications in many fields such as robot navigation, autonomous driving and 3D reconstruction of general environments [19]. Importantly, in contrast to static scenes where the epipolar geometry constraints hold between the corresponding points of different views [17], determining the depth of a moving point from monocular views is an ill-posed problem [3]. This causes standard Structure from Motion techniques [42, 55, 33] to be inadequate in this setup [25].

**Previous work and limitations.**   Many existing approaches for the above problem make simplifying assumptions that limit their applicability to real-world scenarios. Methods based on orthographic camera models and low-rank assumptions use matrix factorization techniques [7, 26], but the orthographic camera assumption might not be realistic and may cause reconstruction errors. Techniques that incorporate depth priors often require lengthy optimization processes in order to make the depth estimates across frames consistent [25, 64]. Other physics-based approaches make assumptions

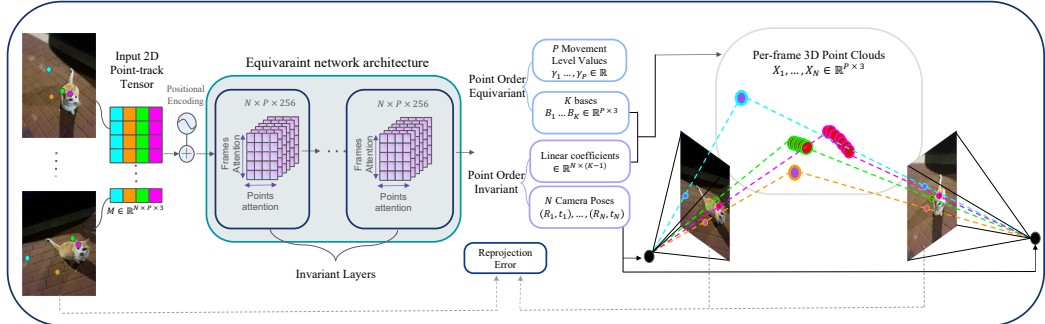

Figure 1: We present TRACKSTO4D, a method for mapping a set of 2D point tracks extracted from casual dynamic videos into their corresponding 3D locations and camera motion. At inference time, our network predicts the dynamic structure and camera motion in a single feed-forward pass. Our network takes as input a set of 2D point tracks (left) and uses several multi-head attention layers while alternating between the time dimension and the track dimension (middle). The network predicts cameras, per-frame 3D points, and per-world point movement value (right). The 3D point internal colors illustrate the predicted 3D movement level values, such that points with high/low 3D motion are presented in red/purple colors respectively. These outputs are used to reproject the predicted points into the frames for calculating the reprojection error losses. See details in the text. The reader is encouraged to watch the supplementary video visualizations.

about rigid bones [60, 58] or isometric deformable surfaces [37] and typically involve complex, slow optimization per video. In addition, they may require foreground-background segmentation of the moving content, which is not always easily obtained. Alternatively, some methods are specifically tailored to certain object classes like humans [52], restricting their domain to those limited cases. Consequently, these prior methods are either not directly applicable to casual videos, or require long optimization time per video.

**Our approach.** We propose TRACKSTO4D,[1] a novel approach for fast reconstruction of sparse dynamic 3D point clouds and camera poses from casual videos. Our main idea is to train a neural network on multiple videos to learn the mapping from the input image sequence to a sequence of the scene's 3D point clouds and camera poses. After training, the trained network can be efficiently applied to new image sequences using a single feed-forward pass, avoiding costly optimization.

To enhance the method's ability to generalize across different types of videos and scenes, we made a crucial design choice: our approach processes point track tensors as input, rather than operating directly on the image sequence. Specifically, each entry $(n, p)$ in these tensors represents the 2D position of a tracked point $p$ in a specific video frame $n$ [7]. Our main insight is that point track tensors may exhibit more common motion patterns across casual video domains compared to image pixels. In other words, we argue that processing the raw point track data rather than scene-specific pixels or features may enable learning class and scene-agnostic internal feature representations for improved generalization. Importantly, recent advances in point tracking [11, 20] enable efficiently inferring these point tracks from casual videos using pre-trained models. These two properties make point track matrices an attractive input for our learning method.

Following this design choice, we design our architecture according to two principles: (1) process point track tensors, which have a unique structure, and (2) encode meaningful prior knowledge about the reconstruction problem, as the problem is ill-posed in general. In the following, we address these desired properties.

First, we design a network architecture that can effectively and efficiently handle point track inputs. To do that, we propose a novel layer design that takes into account the symmetries of the problem: the mapping we aim to learn, from point track matrices to 3D point clouds and camera poses, preserves two natural symmetries: (i) the points being tracked can be arbitrarily permuted without affecting the problem; (ii) the frames containing these points exhibit temporal structure, adhering to an approximate time-translation symmetry. Following the Geometric Deep Learning paradigm [8], we build upon recent theoretical advances in equivariant learning [30] and integrate these two symmetries into our network architecture using dedicated attention and positional encoding mechanisms.

---

[1]4D since we have three Euclidean coordinates with an additional time coordinate

Second, a key challenge in predicting 3D dynamic motion and camera poses from 2D point tracks is that this problem is inherently ill-posed without additional constraints [3]. To address this, we integrate a low-rank movement assumption into our architecture, following the seminal work of [7] which constrained output point clouds to be linear combinations of basis elements. Specifically, given an input point track tensor, our architecture equivariantly predicts a small set of input-specific basis elements. The output point clouds at each time frame are then defined as a linear combination of these basis elements, with the coefficients also predicted by the network. Notably, the first basis is assumed to fully represent the 3D static points in the video, while the remaining basis elements capture the 3D dynamic deviations. This structure effectively restricts the predicted point clouds to have a more specific form, making the problem more constrained.

Our network is trained on a dataset of extracted point track matrices [20] from raw videos without any 3D supervision by simply minimizing the reprojection errors, aiming to predict output point clouds that, after undergoing a perspective projection, will return the original 2D point tracks. In our experiments, TRACKSTO4D is trained on the Common Pets dataset [44]. We evaluate our method on test data with GT camera poses and GT depth information for point tracks, and demonstrate that it produces comparable results to state-of-the-art methods, while having a significantly shorter inference time by up to $95\%$. In addition, we show the method's ability to generalize to out-of-domain videos.

**Contributions.** In summary, our contributions are (1) A novel modeling of the dynamic reconstruction problem via learning on point tracks without 3D supervision; (2) A novel deep learning architecture incorporating two key principles: accounting for the symmetry of the data and encoding low-rank structure in the predicted point clouds (3) Experiments demonstrating extremely fast inference time compared to baselines, accurate results, and strong generalization across other categories.

## 2 Method

**Problem formulation.** Given a video of $N$ frames, let $M \in \mathbb{R}^{N \times P \times 3}$ be a pre-extracted $2D$ point tracks tensor (Fig. 1, left side). This tensor represents the two-dimensional information about a set of $P$ world points that are tracked throughout the video. Each element in the tensor, $M_{i,j,:}$, stores three values: $(x, y, o)$ where $x, y \in \mathbb{R}$ are respectively the observed horizontal and vertical locations of point $j$ in frame $i$, and $o \in \{0, 1\}$ indicates whether point $j$ is observed in frame $i$ or not. Our goal is to train a deep neural network to map the input point tracks tensor $M$ into a set of per-frame camera poses $\{R_i(M), \mathbf{t}_i(M)\}_{i=1}^N$ and per-frame 3D points $\{X_i(M)\}_{i=1}^N$, where $R_i(M) \in \mathbb{SO}(3), \mathbf{t}_i(M) \in \mathbb{R}^3, X_i(M) \in \mathbb{R}^{P \times 3}$ (Fig. 1, right side).

**Overview of our approach.** Our method receives $M \in \mathbb{R}^{N \times P \times 3}$ as input. This tensor is being processed by a neural architecture composed of multi-head attention layers where the attention is applied in an alternating fashion on the $P$ and the $N$ dimensions in each layer. These layers are defined in Sec. 2.1. After a composition of several such layers, the network uses the resulting features in $\mathbb{R}^{N \times P \times d}$ to predict $N$ camera poses in $\mathbb{SO}^3 \times \mathbb{R}^3$ and $N$ point clouds in $\mathbb{R}^{N \times P \times 3}$. These $N$ point clouds are parameterized as a linear combination of $K$ input specific point cloud bases $B_1(M), \ldots B_K(M) \in \mathbb{R}^{P \times 3}$. This is discussed in detail in Sec. 2.2. Our network is trained in an unsupervised way on a dataset of videos by minimizing the reprojection error and other regularization losses (Sec. 2.3) that are used to update the model parameters. Our pipeline is illustrated in Fig. 1

### 2.1 Equivariant layers for point track tensors

Following the geometric deep learning paradigm, our goal is to design a neural architecture that respects the underlying symmetries and structure of the data.

**Symmetry analysis.** Our input is a tensor $M \in \mathbb{R}^{N \times P \times 3}$ representing a sequence of $N$ frames each with $P$ point coordinates. This structure gives rise to two key symmetries: First, the order of

the $P$ points within each frame does not matter - in other words, permuting this axis results in an equivalent problem [30, 39, 63] [2].

Formally, this axis has a permutation symmetry $S_P$ where $S_P$ is the symmetric group on $P$ elements. Second, along the temporal $N$ axis, we have an approximate translation symmetry arising from the ordered video sequence. This means that shifting the time frames is required to result in the same shift in our output. We model this with a cyclic group $C_N$ of order $N$. Both symmetries are illustrated in Fig. 2. We note that while the cyclic group assumption may not be entirely accurate, we still find it useful as it helps us to derive appropriate parametric layers for our data, similar to how the convolutional layer is derived for data with translational symmetries such as images. Taken together, the full symmetry group of the input space is the direct product $\mathcal{G} = C_N \times S_P$ combining these time and point permutation symmetries, acting on $\mathbb{R}^{N \times P \times 3}$ by $((t, \sigma) \cdot M)_{n,p,j} = M_{t^{-1}(n), \sigma^{-1}(p), j}$ for $(t, \sigma) \in \mathcal{G}$ [3]. Next, we will design an architecture equivariant to $\mathcal{G}$, to ensure that the model takes into account the symmetries above.

**Linear equivariant layers.** Point track tensors can be viewed as a collection of $N$ individual point tracks, each of which exhibits translational symmetry. The scenario where an object comprises a set of elements with their own symmetry group, such as a set of images or graphs, was previously explored in [30]. In that work, the authors characterized the general linear equivariant layer structure in such cases, termed the Deep Sets for Symmetric Elements (DSS) layer. Building on the DSS approach, our basic linear equivariant layer for the point track tensors $M$ would take the form:

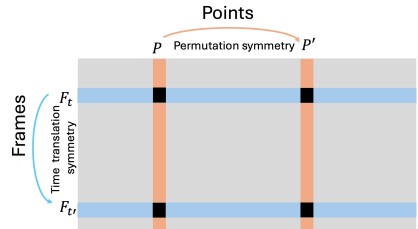

Figure 2: The symmetry structure of our problem. Frames (vertical) have time translation symmetry while points (horizontal) have set permutation symmetry.

$$F(M)_{:,j} = L_1(M_{:,j}) + \sum_{j'=1}^{P} L_2(M_{:,j'}) \quad (1)$$

where $L_i$ are linear translation equivariant function (i.e. convolutions), $M_{:,j} \in \mathbb{R}^{N \times d}$ are the columns of $M$ representing all the inputs for a specific tracked point, $F(M)_{:,j} \in \mathbb{R}^{N \times d'}$ is the output column and $d, d'$ are the input and output feature channels respectively. To construct a neural network, these layers can be interleaved with pointwise nonlinearities, similar to basic convolutional neural networks.

**Implementation via transformers and positional encoding.** While the linear layer design is reasonable, it may not be the optimal choice. To enhance the model, we design a new layer whose structure follows Equation (1), but incorporates nonlinear layers in the form of transformers [51]. We note that the idea of using linear layers as inspiration for non-linear layers aligns with common practices in geometric deep learning, as described in several previous works [30, 4, 14].

Specifically, our layer $F$ is formulated similarly to Equation (1), but instead of convolutions ($L_i$) and summations, it utilizes two self-attention mechanisms and suitable temporal positional encoding across the $N$ dimension. Formally, our basic layer $F : \mathbb{R}^{N \times P \times d} \to \mathbb{R}^{N \times P \times d'}$ is computed via four steps, which are described below:

$$\bar{\mathbf{q}}_{ij} = \bar{W}^Q M_{ij}, \ \bar{\mathbf{k}}_{ij} = \bar{W}^K M_{ij}, \ \bar{\mathbf{v}}_{ij} = \bar{W}^V M_{ij} \Rightarrow \bar{M}_{ij} = \sum_{i'=1}^{N} \frac{\exp(\bar{\mathbf{q}}_{ij} \cdot \bar{\mathbf{k}}_{i'j})}{\sum_{l=1}^{N} \exp(\bar{\mathbf{q}}_{ij} \cdot \bar{\mathbf{k}}_{lj})} \bar{\mathbf{v}}_{i'j} \quad (2)$$

$$\mathbf{q}_{ij} = W^Q \bar{M}_{ij}, \ \mathbf{k}_{ij} = W^K \bar{M}_{ij}, \ \mathbf{v}_{ij} = W^V \bar{M}_{ij} \Rightarrow F(M)_{ij} = \sum_{j'=1}^{P} \frac{\exp(\mathbf{q}_{ij} \cdot \mathbf{k}_{ij'})}{\sum_{l=1}^{P} \exp(\mathbf{q}_{ij} \cdot \mathbf{k}_{il})} \mathbf{v}_{ij'} \quad (3)$$

Here, $M_{i,j} \in \mathbb{R}^d$ are the features associated with the $j$-th point in the $i$-th frame. The attention mechanism defined in the first equation above (2) is augmented with standard temporal positional

---

[2]Our work builds upon the pioneering works PointNet [39] and Deep Sets [63] that studied permutations symmetries in general and for point clouds in particular. As we shall see next, in our case, the symmetry group is a bit more complicated and fits nicely in the setup of [30].

[3]This is different from the symmetry group studied in [32], where the temporal structure was not exploited.

encoding in the first layer and replaces the translation equivariant function $L_i$ applied to the columns of $M$ (Eq.(1)). The attention in the second equation (3) implements the set aggregation (summation) (also in Eq.(1)) applied to the rows of $M$. As commonly done, we use transformers with 16 attention heads [51].

## 2.2 Constraining 3D motion and camera poses via low-rank assumption

Given our 2D tracks, we aim to characterize the motion of the points by decomposing them into the global camera motion and the 3D motion of objects in the scene. The 2D motion of static scene points provides useful constraints for estimating the camera motion. However, as previously mentioned, predicting camera and dynamic 3D motion solely from 2D motion is an ill-posed problem without additional constraints [3]. We tackle this challenge by adding two mechanisms to our architecture: (1) low-rank movement assumption; and (2) specific modeling of the static scene for camera estimation.

**Low-rank movement assumption.** First, motivated by classical orthographic Non-Rigid Structure from Motion [7], we constrain the output points to be formulated by a linear combination of input-specific basis elements. Specifically, given the input 2D point tracks, $M \in \mathbb{R}^{N \times P \times 3}$, our network predicts $K$ point clouds: $B_1(M), \ldots, B_K(M) \in \mathbb{R}^{P \times 3}$ and $N(K-1)$ linear coefficients, $\{c_{1k}(M)\}_{k=2}^K, \ldots \{c_{Nk}(M)\}_{k=2}^K$, such that

$$X_i(M) = B_1(M) + \sum_{k=2}^K c_{ik}(M) B_k(M) \qquad (4)$$

where $X_i(M) \in \mathbb{R}^{P \times 3}$ is the 3D point cloud at frame $i$. The point clouds and coefficients are computed by taking the output of the last equivariant layer as defined in the previous section and applying invariant aggregations on the respective dimension resulting in equivariant and invariant outputs. See more details in the appendix. We note that we deliberately chose the coefficient of $B_1(M)$ to be the constant 1, the reason is explained in the next paragraph.

**Specific modeling of the static scene for camera estimation.** Frequently, casual video data of dynamic scenes contains many static regions, which can be used to determine camera poses [66]. We leverage this observation by treating the first basis element $B_1(M) \in \mathbb{R}^{P \times 3}$ as a static approximation for all scene points and encourage $B_1(M)$ as well as the output camera poses to explain the 2D observations according to this approximation using a "static" reprojection loss ($\mathcal{L}_{\text{Static}}$, defined in the next section). We note, however, that a static point cloud is not likely to produce low reprojection errors for the non-static components, thus the reprojection error necessitates robustness to substantial errors from the non-static elements. To address this, our network predicts (equivariantly) $P$ motion level values $\gamma_1(M), \ldots, \gamma_P(M) \in \mathbb{R}_+$, one for each point in our dynamic point cloud, which we use to weight the reprojection errors from $B_1(M)$. The main idea is to give less weight to non-static points so that the static projection loss can disregard them. Specifically, inspired by [64], each $\gamma_i(M)$ defines a Cauchy distribution that models the reprojection errors for its associated world point, such that a world point with higher $\gamma$ is expected to produce a wider error distribution. Empirically, as noted by [64], the Cauchy distribution tends to be more robust for modeling reprojection error uncertainties compared to Gaussian noise modeling [22]. Then, $\mathcal{L}_{\text{Static}}$, minimizes the negative log-likelihood under this assumption. See details in Sec. 2.3.

**Comparison to supervised learning setups.** A central aspect of our method is addressing the ill-posed nature of this problem through an architecture that heavily regularizes the output dynamic point clouds, enabling training without supervision. Alternatively, one could use video data with ground truth depth for supervised training, potentially allowing the output to have a more general form. Unfortunately, such supervision is both challenging to acquire and may lead to poor generalization on novel motion patterns.

## 2.3 Training and losses

**Model outputs.** Given the input 2D point tracks $M \in \mathbb{R}^{N \times P \times 3}$, our network produces outputs as a function of $M$: linear bases and coefficients,

$B_1(M), \ldots, B_K(M) \in \mathbb{R}^{P \times 3}, \{c_{1k}(M)\}_{k=2}^K, \ldots, \{c_{Nk}(M)\}_{k=2}^K \in \mathbb{R}$ which define a dynamic point cloud $X_1(M), \cdots, X_N(M) \in \mathbb{R}^{P \times 3}, \gamma_1(M), \ldots, \gamma_P(M) \in \mathbb{R}_+$ movement level values, and $(R_1(M), \mathbf{t}_1(M)), \ldots, (R_N(M), \mathbf{t}_N(M)) \in SO(3) \times \mathbb{R}^3$ camera poses.

We use these network outputs to define a self-supervised loss function with respect to the current network weights and $M$ which is defined by:

$$\mathcal{L} = \lambda_{\text{Reproject}} \mathcal{L}_{\text{Reproject}} + \lambda_{\text{Static}} \mathcal{L}_{\text{Static}} + \lambda_{\text{Negative}} \mathcal{L}_{\text{Negative}} + \lambda_{\text{Sparse}} \mathcal{L}_{\text{Sparse}} \tag{5}$$

**Reprojection loss.** The reprojection loss encourages the consistency between the output 3D point clouds and camera poses, to the input 2D observations:

$$\mathcal{L}_{\text{Reproject}} = \frac{1}{\sum_{i=1}^N \sum_{j=1}^P M_{ij}^o} \sum_{i=1}^N \sum_{j=1}^P M_{ij}^o \mathcal{R}(X_{ij}, R_i, \mathbf{t}_i, M_{ij}^{xy}) \tag{6}$$

where $\mathcal{R}(X_{ij}, R_i, \mathbf{t}_i, M_{ij}^{xy})$ is the reprojection error when projecting the point $X_{ij}$ with the camera pose $(R_i, \mathbf{t}_i)$ with respect to the measured point $M_{ij}^{xy}$:

$$\mathcal{R}(X_{ij}, R_i, \mathbf{t}_i, M_{ij}^{xy}) = \left\| \frac{(R_i^T(\mathbf{X}_{ij} - \mathbf{t}_i))_{1,2}}{(R_i^T(\mathbf{X}_{ij} - \mathbf{t}_i))_3} - M_{ij}^{xy} \right\| \tag{7}$$

**Static loss.** As discussed in Sec. 2.2, to better constrain the camera poses, the first predicted basis element $B_1(M) \in \mathbb{R}^{P \times 3}$ defines a static (fixed in time) point cloud approximation. Our network also predicts a movement coefficient $\gamma_j(M)$ for each world point that defines a zero-mean Cauchy distribution. Given $\gamma_j$ and the reprojection error $r_{ij} = \mathcal{R}(B_{1j}, R_i, \mathbf{t}_i, M_{ij}^{xy})^4$ of the $j^{th}$ point of $B_1$ that is projected by the $i^{th}$ camera, the negative log-likelihood of $r_{ij}$ distributed according to the $\gamma_j$-zero-mean Cauchy distribution is proportional to:

$$\mathcal{C}(r_{ij}, \gamma_j) = \log \left( \gamma_j + \frac{r_{ij}^2}{\gamma_j} \right) \tag{8}$$

Note, that this loss reduces the contribution of the reprojection errors for points with high $\gamma$, but also encourages $\gamma$ to be small, i.e. encouraging the static point cloud to represent the dynamic scene when possible. Our static loss is the mean negative log-likelihood over all observed points in all frames:

$$\mathcal{L}_{\text{Static}} = \frac{1}{\sum_{i=1}^N \sum_{j=1}^P M_{ij}^o} \sum_{i=1}^N \sum_{j=1}^P M_{ij}^o \mathcal{C}(\mathcal{R}(B_{1j}, R_i, \mathbf{t}_i, M_{ij}^{xy}), \gamma_j) \tag{9}$$

**Regularization losses.** As in [32] we regularize the observed points to be in front of the camera by:

$$\mathcal{L}_{\text{Negative}} = - \sum_{i=1}^N \sum_{j=1}^P M_{ij}^o \, \text{Min}(R_i^T(\mathbf{X}_{ij} - \mathbf{t}_i)_3, 0) \tag{10}$$

We further find it beneficial to regularize the deviation from the static approximation $B_1$ to be sparse for static points, i.e. points with low $\gamma$ values:

$$\mathcal{L}_{\text{Sparse}} = \frac{1}{P(K-1)} \sum_{k=2}^K \sum_{j=1}^P \frac{1}{3\gamma_j} (|B_{kj1}| + |B_{kj2}| + |B_{kj3}|) \tag{11}$$

where $\gamma$ is detached from the gradient calculation for this loss.

## 3 Experiments

In this section, we conduct experiments to verify our proposed network's performance on real-world casual videos. We began by training the network on specific domains and then evaluated its accuracy and running time on unseen videos from both, training and unseen domains.

---

[4] We denote the $j^{th}$ 3D point of $B_k \in \mathbb{R}^{P \times 3}$ by $B_{kj} \in \mathbb{R}^3$. The 3 elements of this point are denoted by $B_{kj1}, B_{kj2}, B_{kj3} \in \mathbb{R}$ (see (11)).

Table 1: **Pet evaluation**. **Top**: Baseline method results for structure or camera estimation (or both). **Bottom**: Our results with several configurations. (C),(D), or (CD) respectively indicate the object categories in the training set: cats, dogs, or both. BA and FT respectively indicate a post-processing of Bundle Adjustment or fine-tuning.

| | Abs Rel ↓ | | $\delta < 1.25$ ↑ | | $\delta < 1.25^2$ ↑ | | $\delta < 1.25^3$ ↑ | | ATE ↓ (mm) | RPE$_{Trans}$ ↓ (mm) | RPE$_{Rot}$ ↓ (deg) | Time (min) |
|---|---|---|---|---|---|---|---|---|---|---|---|---|
| | Dyn. | All | Dyn. | All | Dyn. | All | Dyn. | All | | | | |
| D-SLAM [48] | - | - | - | - | - | - | - | - | 5.08 | 3.60 | 0.20 | 0.16 |
| ParticleSFM [66] | - | - | - | - | - | - | - | - | 12.79 | 6.95 | 0.51 | 11.00 |
| RCVD [25] | 0.40 | 3.6E+07 | 0.43 | 0.72 | 0.75 | 0.90 | 0.92 | 0.96 | 43.95 | 25.77 | 2.31 | 20.00 |
| CasualSAM [64] | **0.09** | **0.06** | **0.93** | **0.97** | 0.99 | **0.99** | **1.00** | **1.00** | 6.90 | 3.95 | 0.22 | 130 |
| MiDaS [5] | 0.16 | 6.2E+04 | 0.78 | 0.71 | 0.97 | 0.88 | **1.00** | 0.93 | - | - | - | **0.15** |
| Ours (C) | 0.11 | 0.08 | 0.88 | 0.92 | 0.99 | 0.98 | **1.00** | **1.00** | 8.96 | 3.79 | 0.23 | **0.15** |
| Ours (C)+BA | 0.11 | 0.08 | 0.88 | 0.92 | 0.99 | 0.98 | **1.00** | **1.00** | 4.22 | 2.86 | 0.17 | **0.15** |
| Ours (C)+FT | **0.09** | **0.06** | 0.90 | 0.96 | **1.00** | **0.99** | **1.00** | **1.00** | 4.00 | **2.74** | 0.16 | 4.86 |
| Ours (D) | 0.12 | 0.08 | 0.85 | 0.91 | 0.99 | 0.99 | **1.00** | **1.00** | 8.03 | 3.54 | 0.23 | **0.15** |
| Ours (D)+BA | 0.12 | 0.08 | 0.85 | 0.91 | 0.99 | **0.99** | **1.00** | **1.00** | 4.19 | 2.83 | 0.17 | **0.15** |
| Ours (D)+FT | **0.09** | **0.06** | 0.88 | 0.96 | **1.00** | **0.99** | **1.00** | **1.00** | 3.98 | **2.74** | 0.16 | 4.86 |
| Ours (CD) | 0.12 | 0.08 | 0.85 | 0.91 | 0.98 | 0.98 | **1.00** | **1.00** | 8.11 | 3.68 | 0.24 | **0.15** |
| Ours (CD)+BA | 0.12 | 0.08 | 0.85 | 0.91 | 0.98 | 0.98 | **1.00** | **1.00** | 4.21 | 2.86 | 0.17 | **0.15** |
| Ours (CD)+FT | **0.09** | **0.06** | 0.90 | 0.96 | **1.00** | **0.99** | **1.00** | **1.00** | 3.98 | **2.74** | 0.16 | 4.86 |

Table 2: **Out-of-training-domain evaluation** . Evaluation metrics on monocular videos from [62]. The table has the same structure as Tab. 1.

| | Abs Rel ↓ | | $\delta < 1.25$ ↑ | | $\delta < 1.25^2$ ↑ | | $\delta < 1.25^3$ ↑ | | ATE ↓ (mm) | RPE$_{Trans}$ ↓ (mm) | RPE$_{Rot}$ ↓ (deg) | Time (min) |
|---|---|---|---|---|---|---|---|---|---|---|---|---|
| | Dyn. | All | Dyn. | All | Dyn. | All | Dyn. | All | | | | |
| D-SLAM [48] | - | - | - | - | - | - | - | - | 7.96 | 10.91 | 0.07 | 0.18 |
| ParticleSFM [66] | - | - | - | - | - | - | - | - | 26.66 | 23.83 | 0.20 | 2.13 |
| RCVD [25] | 0.19 | 2.6E+05 | 0.69 | 0.75 | 0.95 | 0.95 | 0.96 | 0.98 | 160 | 320 | 3.43 | 7.00 |
| CasualSAM [64] | **0.05** | **0.03** | 0.95 | **0.99** | **0.99** | **1.00** | **1.00** | **1.00** | **7.81** | **10.09** | **0.06** | 22.00 |
| MiDaS [5] | 2.8E+04 | 2.7E+05 | 0.59 | 0.58 | 0.73 | 0.72 | 0.83 | 0.80 | - | - | - | **0.02** |
| Ours (C) | 0.08 | 0.06 | 0.89 | 0.95 | **0.99** | 0.99 | 0.99 | **1.00** | 32.06 | 47.99 | 0.45 | 0.04 |
| Ours (C)+BA | 0.08 | 0.06 | 0.89 | 0.95 | **0.99** | 0.99 | 0.99 | **1.00** | 8.67 | 12.36 | 0.08 | 0.04 |
| Ours (C)+FT | 0.07 | **0.03** | 0.94 | 0.98 | **0.99** | **1.00** | **1.00** | **1.00** | 7.98 | 11.64 | 0.08 | 0.59 |
| Ours (D) | 0.08 | 0.07 | 0.92 | 0.93 | **0.99** | 0.98 | 0.99 | **1.00** | 33.77 | 51.64 | 0.61 | 0.04 |
| Ours (D)+BA | 0.08 | 0.07 | 0.92 | 0.93 | **0.99** | 0.98 | 0.99 | **1.00** | 8.40 | 12.06 | 0.08 | 0.04 |
| Ours (D)+FT | **0.05** | **0.03** | **0.97** | **0.99** | **0.99** | **1.00** | 0.99 | **1.00** | 8.15 | 11.88 | 0.09 | 0.59 |
| Ours (CD) | 0.10 | 0.08 | 0.93 | 0.94 | **0.99** | 0.99 | **1.00** | **1.00** | 36.17 | 53.94 | 0.67 | 0.04 |
| Ours (CD)+BA | 0.10 | 0.08 | 0.93 | 0.94 | **0.99** | 0.99 | **1.00** | **1.00** | 8.62 | 12.49 | 0.08 | 0.04 |
| Ours (CD)+FT | 0.06 | **0.03** | **0.97** | **0.99** | **0.99** | **1.00** | 0.99 | **1.00** | 8.04 | 11.84 | 0.09 | 0.59 |

**Training procedure.** We trained our network on the cat and dog partitions from the COP3D dataset [44], which contains a diverse set of casual real-world videos of pets. Our networks were trained from scratch three times to test our generalization capability between semantic categories: once on the cat partition, once on the dog partition, and once on both partitions combined. Training technical details are provided in the appendix. We use $K = 12$ bases in all our experiments (Sec. 2.2).

**Evaluation data.** To assess our framework's performance on pet videos, we curated a new dataset[5] consisting of 21 casual videos of dogs and cats, each video containing 50 frames. These videos were captured using an RGBD (RGB-Depth) sensor. The depth maps were used as ground truth for evaluating the reconstructed structure. We extracted the cameras by running COLMAP on the images while masking out the pet areas with dilatated masks provided by [71]. The cameras were scaled to millimeter units using the provided GT depth. Note that our network did not see this test data during training and it was not used to tune our hyperparameters.

Additionally, to evaluate our method on out-of-domain evaluation data, we used the Nvidia Dynamic Scenes Dataset [62]. Specifically, while our network was trained on pet videos, this dataset contains other dynamic object types, e.g. human, balloon, truck, and umbrella, with a different camera motion type and a variety of motion profiles. The dataset contains 8 dynamic scenes which are captured by 12 synchronized cameras, enabling accurate depth estimation which is treated as GT for evaluating monocular depth estimation. The ground truth camera poses were calculated by [42] with the synchronized multiview camera rig and the ground truth dynamic masks. Similarly to [27] we

---

[5]While the COP3D dataset provides cameras that were extracted by COLMAP [42], we note that this evaluation data is insufficient in our case. This is because the dynamic structure was captured as well in part of their reconstruction which indicates that its reconstruction might not be accurate. Furthermore, the coordinates system units of these reconstructions are unknown. Finally, this dataset does not have any depth map information for evaluating the dynamic structure.

Table 3: **Ablation study.** The contribution of different parts from our method. See details in the text.

| | Abs Rel ↓ | | $\delta < 1.25$ ↑ | | $\delta < 1.25^2$ ↑ | | $\delta < 1.25^3$ ↑ | | Rep.(pix.) ↓ | | ATE ↓ | RPE Trans ↓ | RPE Rot ↓ |
| | Dyn. | All | Dyn. | All | Dyn. | All | Dyn. | All | Dyn. | All | (mm) | (mm) | (deg) |
|---|---|---|---|---|---|---|---|---|---|---|---|---|---|
| Set of Sets | 0.27 | 0.15 | 0.60 | 0.77 | 0.87 | 0.94 | 0.97 | 0.99 | 9.86 | 5.33 | 16.87 | 5.53 | 0.39 |
| No $\mathcal{L}_{\text{Static}}$ | 0.77 | 0.36 | 0.25 | 0.46 | 0.48 | 0.70 | 0.68 | 0.82 | **1.00** | **0.86** | 96.20 | 29.86 | 0.99 |
| No $\gamma$ | 0.22 | 0.16 | 0.66 | 0.73 | 0.93 | 0.91 | 0.99 | 0.97 | 4.54 | 2.41 | 13.91 | 4.86 | 0.29 |
| K=30 | 0.14 | 0.09 | 0.81 | 0.90 | 0.97 | **0.98** | 0.99 | 0.99 | 4.88 | 2.78 | 9.39 | **3.68** | **0.23** |
| K=2 | **0.11** | **0.08** | **0.88** | 0.91 | 0.98 | **0.98** | **1.00** | **1.00** | 8.58 | 3.56 | 9.31 | 3.86 | 0.25 |
| DSS | 1.65 | 0.58 | 0.19 | 0.35 | 0.34 | 0.60 | 0.47 | 0.74 | 63.75 | 70.60 | 34.90 | 22.63 | 1.64 |
| No $\mathcal{L}_{\text{Sparse}}$ | 0.17 | 0.13 | 0.79 | 0.80 | 0.95 | 0.94 | **1.00** | 0.99 | 4.57 | 2.73 | 11.79 | 7.99 | 0.55 |
| Full | **0.11** | **0.08** | **0.88** | **0.92** | **0.99** | **0.98** | **1.00** | **1.00** | 3.98 | 1.97 | **8.96** | 3.79 | **0.23** |

simulated 8 monocular dynamic video sequences using the camera rig, each with 24 frames, and used them for evaluation.

**Evaluation results.** Qualitative visualizations are presented in Fig. 3.[6] We also show a visualization of the movement level values, $\gamma$ in Fig. 4. For comparisons, we chose state-of-the-art methods that as our method, can be applied to raw casual videos that were captured by standard pinhole camera models and do not need any static or dynamic segmentation. We evaluated both, the camera poses and the structure accuracies. Comparison results for the pet-test-set and out-of-domain dataset are presented in Tables 1 and 2 respectively. The camera poses are evaluated compared to the GT, using the Absolute Translation Error(ATE), the Relative Translation Error(RTE), and the Relative Rotation Error(RRE) metrics after coordinates system alignment. We compare three training configurations of our method of training only on cats, only on dogs, and on both. As can be seen in the tables, the results are similar in all 3 cases. Our output camera poses as inferred by the network are already accurate and outperform some of the prior methods. We further show the results of our method after a single and short round of Bundle Adjustment, which makes our method better than all baselines on the pet sequences, and comparable on the out-of-domain cases.

Importantly, Tables 1 and 2 also compare the method's runtime. It can be seen that our method, even with bundle adjustment, is the fastest camera prediction method. Note that our method's runtimes include the point tracking time that is performed by [20] as a pre-process. In the appendix (Tab. 8), we present tracking time versus inference time, showing that most of the time is spent on tracking, while our inference is very fast. Tables 1 and 2 also show structure evaluation with the depth evaluation metrics [12] on the sampled point tracks. They demonstrate that our inferred structure is almost comparable to the state-of-the-art [64] while taking significantly shorter running times (a few seconds for our method versus more than two hours for [64] on pet videos). Short (0.6-5 minutes), per-sequence fine-tuning makes our method's accuracy comparable to [64]. In the appendix, we present per-scene output accuracy, demonstrating our ability to generalize across different speed profiles. In terms of running time, our method is a bit slower than MiDaS [5] which only provides depth maps without cameras, but achieves much better results[7]. We note that in contrast to the other methods that predict the dynamic depth, ours does not use any depth-from-single-image prior.

**Ablation study** To evaluate the contribution of our different method parts we run an ablation study which is presented in Tab. 3. In this study, the training was always done on the cat partition from COP3D and evaluated on our test data which contains dogs and cats. First, we performed an ablation study on our transformer architecture by taking the architecture suggested by [32] ("Set of Sets") or the DSS architecture that uses only linear layers [30] ("DSS"). As the table shows our architecture ("Full") achieved significantly better results. To test the losses in our framework, we also evaluated the following: (1) ignoring the $\gamma$ outputs and using regular reprojection error on $B_1$ for all points ("No $\gamma$"); (2) removing our sparsity loss ("No $\mathcal{L}_{\text{Sparse}}$"); and (3) removing the static loss ("No $\mathcal{L}_{\text{Static}}$"). In all cases, the error increased whereas in the later one, the results became much worse. We further ablate the choice of $K = 12$ as the number of linear bases, by trying 2 extreme numbers of $K = 30, K = 2$ (we saw no significant differences when we used nearby choices such as $K = 11$). As can be seen in the table, when $K = 30$ the output is not regularized enough and produces a higher

---

[6]The reader is encouraged to watch the supplementary videos for better 4D perception.

[7]Our comparisons were made using MiDaS 3.1, specifically the dpt_beit_large_512 version (Birkl et al. 2023). This is an improved version of MiDaS that utilizes the DPT architecture, representing a more current state of the method.

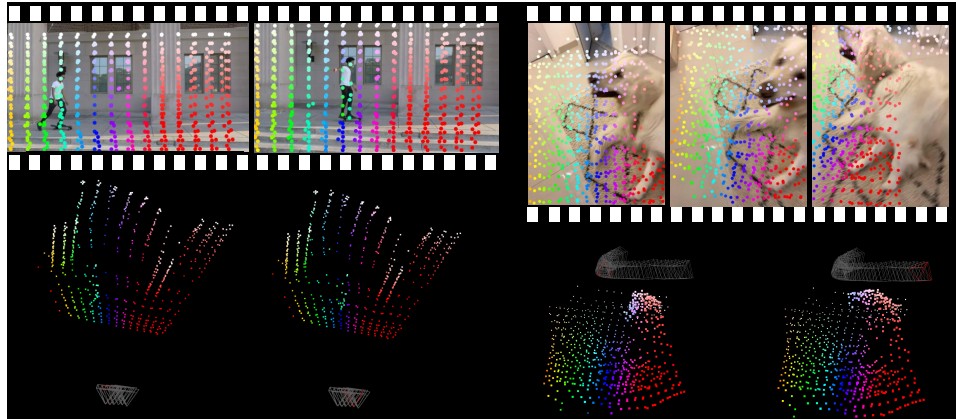

Figure 3: **Qualitative Results. Top**. Frames from 2 different test video sequences with point tracks marked with corresponding colors. **Bottom**. A 3D visualization of our method's outputs, from two time stamps. The camera trajectory is present as gray frustums, whereas the current camera is marked in red. The reconstructed 3D scene points are presented in corresponding colors to the input tracks on the top. The scene is observed from the same viewpoint, enabling the visualization of the dynamic reconstructed structure.

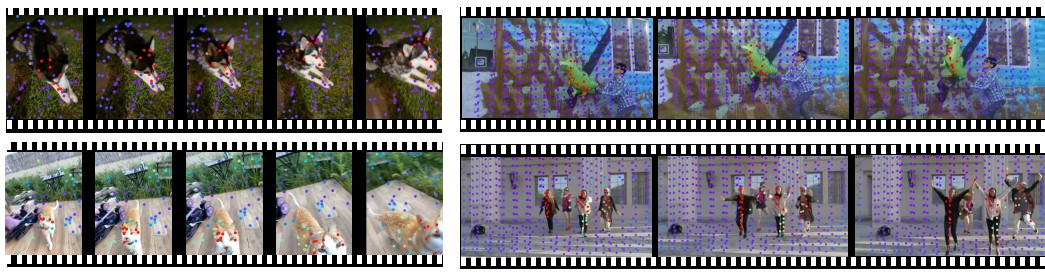

Figure 4: $\gamma$ **Visualization.** We show a visualization of the $\gamma$ outputs of our network that are described in Sec. 2.2. In each video sequence, we show the input tracks, where each color visualizes its movement level value, $\gamma$. Purple marks static points with low $\gamma$ whereas red marks dynamic points with high $\gamma$. Note, that our network did not get any direct supervision for these values, but only the raw point tracks predictions from [20]. The $\gamma$ visualizations for cats were produced by the model that was only trained on dogs and vice versa. We note that our model generalizes well to out-of-domain (non-pet) cases as well.

depth error for the dynamic part. For $K = 2$ the depth is regularized but the reprojection error ("Rep. (pix.)") gets higher due to over-regularization. Overall, this study justifies our design choices ("Full").

## 4   Related Work

**Simultaneous Localization and Mapping (SLAM) and Structure from Motion (SfM)**    SfM pipelines seek to recover static 3D structure and camera poses from unordered images.[49, 45, 42, 1, 55]. Learning-free pipelines [42] are effective but require repeated applications of Bundle Adjustment (BA) [50]. [32, 9] presented a method for learning prior from a dataset of multiview image sets, to accelerate SfM pipelines by using equivariant deep networks. Monocular Simultaneous Localization and Mapping (SLAM) methods [33, 34, 13, 61, 6, 54, 67, 69, 46] extract camera poses from video sequences while defining a scene map with keyframes. These methods assume static scenes, fail to produce the cameras in scenes with large portions of dynamic motion, and cannot reproduce dynamic parts of the scene. DROID-SLAM [48] used synthetic data with ground truth 3D supervision for learning to predict camera poses via deep-based BA on keyframes while excluding dynamic objects. ParticleSfM [66] filters out 2D dynamic content for reproducing the cameras in dynamic scenes, using its pre-trained motion prediction network. Both, [48, 66] do not infer the dynamic 3D structure.

**Orthographic Non-Rigid SfM (NRSfM)**    Bregler et al. [7] introduced a factorization method for computing a non-rigid structure and rotation matrices from a point track matrix, by assuming a low

dimensional basis model. While follow-up papers improved accuracies with different regularizations [26, 10, 36, 18] or neural representations [35, 24, 43], the orthographic camera model assumption is in general not valid for casual videos. Furthermore, these methods often assume background subtraction as a preprocessing. Even though a follow-up work proposed factorization solutions for pinhole cameras [16], its sensitivity to noise [19], makes it impractical for casual videos.

**Test-time optimization for dynamic scenes**    Recent methods [29, 25, 64, 65] finetuned the monocular depth estimation from a pre-trained model [41, 40] using optical flow constraints [47], for obtaining consistent dense depth maps for a monocular video. [64] further optimized motion maps for handling scenes with large dynamic motion. [56, 15] use depth from single image estimations, to improve novel view synthesis in dynamic scenes. [28] further optimizes for the unknown camera poses together with the dynamic radiance field optimization. [37, 38] model a single deformable surface from a monocular video by applying isometric constraints. LASR [58], ViSER [59] and BANMo [60] optimize for a dynamic surface by assuming rigid bones and linear blend skinning weights. However, all the above-mentioned methods require per-scene optimization, resulting in slow inference. Recently, [44] presented the Common Pets in 3D (COP3D) dataset that contains casual, in-the-wild videos of pets, and used it to learn priors for novel view synthesis in dynamic scenes.

**Point tracking**    There has been a recent advance in 2D point tracking by learning [20, 11], or optimization [53] techniques. Concurrently, [57] presented a method for jointly performing 2D tracking and 3D lifting, by learning to track with depth supervision while applying an as-rigid-as-possible loss. However, their method cannot predict camera poses or identify static parts directly.

## 5    Conclusions and limitations

We presented TRACKSTO4D, a novel deep-learning framework that directly maps 2D motion tracks from casual videos into their corresponding dynamic structure and camera motion. Our approach features a deep learning architecture that considers the symmetries in the problem with designed intrinsic constraints to handle the ill-posed nature of this problem. Notably, our network was trained using only raw supervision of 2D point tracks extracted by an off-the-shelf method [20] without any 3D supervision. Yet, it implicitly learned to predict camera poses and 3D structures while identifying the dynamic parts. During inference, our method demonstrates significantly faster processing times compared to previous methods while achieving comparable accuracy. Furthermore, our network exhibits strong generalization capabilities, performing well even on semantic categories that were not present in the training data.

**Limitations and future work.**    While our experiments demonstrated that our network is efficient, accurate, and capable of generalizing to unseen video categories, there are several limitations and future work directions that we would like to address. First, our method is limited in handling videos with rapid motion, as it depends on the accuracy of the tracking method used, specifically CoTracker [20], which has limitations in tracking fast movements. We observed that [20] often fails in automotive scenes due to high motion blur, particularly on the road, limiting our ability to perform large-scale evaluations on automotive datasets given the current tracking method performance. We note that any future improvements with point tracking in terms of accuracy and inference time will directly improve our method as well. Second, our method assumes enough motion parallax to constrain the depth values and fails to generate accurate camera poses without it. An interesting direction for future work would be to leverage depth models and 3D tracking techniques, which could potentially improve accuracy, especially in cases with minimal motion parallax. Yet, our current attempts to incorporate MiDaS-inferred depth as an additional input and apply a depth loss relative to MiDaS have not yielded any performance improvements, indicating that this approach needs further exploration. Third, while we found $K = 12$ basis elements to be effective for our evaluation set, balancing complexity reduction and motion representation, we acknowledge this fixed number may not capture all possible scene dynamics. Future work could explore automatically inferring the optimal number of bases per scene. Lastly, our network can handle up to about 1000 point tracks in 50 frames in one inference step when running on a single GPU. A possible extension to handle denser point clouds and longer videos could involve querying point tracks iteratively while maintaining a shared state, but this approach remains to be explored.

**Acknowledgments**  HM is the Robert J. Shillman Fellow, and is supported by the Israel Science Foundation through a personal grant (ISF 264/23) and an equipment grant (ISF 532/23).

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

# A  Supplementary results

## A.1  Video results

We provide supplementary video outputs of several cases from our test set. Each video presents the input video frames with a set of pre-extracted point tracks that are used as input to our network and presented in corresponding colors (left side), and the output cameras and dynamic 3D structure (right side). The output camera trajectory is presented as gray frustums, whereas the current camera is marked in red. The reconstructed 3D scene points are presented in corresponding colors to the input tracks. Note that the outputs presented in the videos were obtained at inference time, with a single feed-forward prediction, without any optimization or fine-tuning, on unseen test cases.

## A.2  Extended quantitative evaluation

**Per sequence quantitative numbers**   Per-sequence and mean quantitative comparisons for our 21 pet test videos are presented in Tab. 9 and Tab. 10. Tables with similar structure for the out-of-domain dataset are presented in Tab. 11 and Tab. 12. Tables 11 and 12 also demonstrate our robustness across various levels of dynamic motion. However, some differences in tolerance to motion patterns are evident. For instance, our depth accuracy on 'Skating' is slightly higher than on 'Truck'.

**Handling tracking errors**   We note that the input point tracks extracted by CoTracker [18] are imperfect and contain noise and outliers. Nevertheless, our model shows significant robustness to these errors. Our model handles imperfect input through two features:

- **Static scene modeling:** Ensures that only 2D motion that can be represented by a camera motion and truly static points is modeled statically. This makes our camera estimation robust to errors while pushing the non-modelled errors to the dynamic part.
- **Dynamic scene modeling:** Using limited basis elements for dynamic parts inherently resists outliers and extreme anomalies.

To quantify this robustness, we conducted tests by adding Gaussian and uniform noise to CoTracker points. The results (see Tab. 4) confirm that our method tolerates significant noise levels, further validating its effectiveness in handling imperfect input data.

**Robustness to tracking method**   Additionally, we tested using point tracks extracted by TAPIR [11] as input at inference, instead of the CoTracker point tracks [20]. This evaluates our method's ability to generalize to a different tracking method at inference than the one used during training. The results are shown in Tab. 5. Although we observed some degradation in accuracy, the results are still good especially after finetunning, demonstrating our method's robustness to different point trackers.

**Synthetic Data Evaluation**   We ran an evaluation on multiple test cases from PointOdyssey [68] and compared them with CasualSAM [64], which is the most accurate baseline. The results are presented in Tab. 6. We observed that our method generalizes well to these cases while taking much less time than CasualSAM and maintaining high accuracy.

**Comparsion with Marigold**   We also add new experiments with the Marigold depth estimation method [21]. The numbers are presented in Tab. 7, demonstrating that our method is more accurate in terms of depth accuracy.

# B  Implementation details

## B.1  Architecture technical details

For learning high frequencies we map each input coordinate to sinusoidal functions as in [31] with $L = 12$. We use 3 pairs of attention layers, each of frames attention followed by point attention. Each point (after sinusoidal functions embedding) is mapped into $\mathbb{R}^{256}$ with a linear layer. Each attention layer is a function of the form $F : \mathbb{R}^{N \times P \times 256} \to \mathbb{R}^{N \times P \times 256}$ (see details above). Each attention uses 16 heads with $K, Q, V \in \mathbb{R}^{(N \text{ or } P) \times 64}$ followed by a fully connected network with 1

Table 4: **Tracking Error**. The table demonstrates our point track error handling and contains four row sections. **1. Ours**: our results when evaluating on the original CoTracker point tracks from the pet test set. **2. $\sigma = 1, 5, 10$**: our results using the original CoTracker point tracks with added Gaussian noise with corresponding standard deviation in pixels. **3. 10,20,50% outliers**: our results using the original CoTracker point tracks where respectively 10,20,50 percent of the tracks are replaced with uniformly sampled pixels. **4. 10,20,50% occlusions**: same as the outlier setup, but the outlier points are marked as occluded by setting $o = 0$ (defined in Sec. 2), which improves outlier handling. Overall we see that our method can robustly handle the noisy CoTracker inputs, and can further tolerate a significant level of synthetically added noise.

| | Abs Rel ↓ | | $\delta < 1.25$ ↑ | | $\delta < 1.25^2$ ↑ | | $\delta < 1.25^3$ ↑ | | ATE ↓ | RPE Trans ↓ | RPE Rot ↓ |
| | Dyn. | All | Dyn. | All | Dyn. | All | Dyn. | All | (mm) | (mm) | (deg) |
|---|---|---|---|---|---|---|---|---|---|---|---|
| Ours | 0.11 | 0.08 | 0.88 | 0.92 | 0.99 | 0.98 | 1.00 | 1.00 | 8.96 | 3.79 | 0.23 |
| $\sigma = 1$ | 0.11 | 0.08 | 0.88 | 0.92 | 0.99 | 0.98 | 1.00 | 1.00 | 9.07 | 3.87 | 0.24 |
| $\sigma = 5$ | 0.11 | 0.08 | 0.88 | 0.92 | 0.99 | 0.98 | 1.00 | 1.00 | 10.96 | 5.05 | 0.30 |
| $\sigma = 10$ | 0.13 | 0.10 | 0.84 | 0.90 | 0.98 | 0.98 | 1.00 | 0.99 | 13.18 | 6.75 | 0.42 |
| 10% outliers | 0.26 | 0.19 | 0.74 | 0.74 | 0.89 | 0.90 | 0.94 | 0.96 | 14.45 | 5.92 | 0.45 |
| 20% outliers | 0.35 | 0.23 | 0.62 | 0.65 | 0.82 | 0.85 | 0.90 | 0.94 | 15.26 | 5.96 | 0.46 |
| 50% outliers | 0.61 | 0.32 | 0.36 | 0.51 | 0.60 | 0.75 | 0.78 | 0.88 | 19.68 | 11.41 | 0.69 |
| 10% occlusions | 0.22 | 0.16 | 0.80 | 0.80 | 0.92 | 0.93 | 0.95 | 0.97 | 11.68 | 4.58 | 0.30 |
| 20% occlusions | 0.30 | 0.20 | 0.70 | 0.71 | 0.85 | 0.89 | 0.92 | 0.95 | 13.15 | 5.04 | 0.35 |
| 50% occlusions | 0.58 | 0.33 | 0.38 | 0.51 | 0.64 | 0.76 | 0.82 | 0.88 | 16.18 | 5.82 | 0.51 |

Table 5: **TAPIR point tracks**. Comparison of using our method with TAPIR [11] point tracks as input at inference time versus using CoTracker [20], which was used for training our method, on the pet test set. As can be seen, our method is robust to the point tracks obtained by TAPIR, and the results are further improved when applying test time finetuning (FT).

| | Abs Rel ↓ | | $\delta < 1.25$ ↑ | | $\delta < 1.25^2$ ↑ | | $\delta < 1.25^3$ ↑ | | ATE ↓ | RPE Trans ↓ | RPE Rot ↓ |
| | Dyn. | All | Dyn. | All | Dyn. | All | Dyn. | All | (mm) | (mm) | (deg) |
|---|---|---|---|---|---|---|---|---|---|---|---|
| Ours (C) CoTracker | 0.11 | 0.08 | 0.88 | 0.92 | 0.99 | 0.98 | 1.00 | 1.00 | 8.96 | 3.79 | 0.23 |
| Ours (C) TAPIR | 0.14 | 0.11 | 0.82 | 0.85 | 0.96 | 0.96 | 0.99 | 0.99 | 11.86 | 5.00 | 0.36 |
| Ours (C) TAPIR + BA | 0.14 | 0.11 | 0.82 | 0.85 | 0.96 | 0.96 | 0.99 | 0.99 | 6.08 | 4.14 | 0.26 |
| Ours (C) TAPIR + FT | 0.09 | 0.06 | 0.88 | 0.94 | 0.99 | 0.99 | 1.00 | 0.99 | 6.66 | 3.88 | 0.22 |

hidden layer of 2048 features. We then average over the rows to get per-point features $P_0 \in \mathbb{R}^{P \times 256}$ and over the columns to get per-frame features $F_0 \in \mathbb{R}^{N \times 256}$. Finally, we map $P_0$ to per-point outputs $P_1 \in \mathbb{R}^{P \times (3K+1)}$ ( K basis points and $\gamma$) with a linear layer, and $F_0$ into per-camera outputs $F_1 \in \mathbb{R}^{N \times (6+3+K-1)}$ (6 for the rotation parameters [70], 3 for the camera center, and $K - 1$ linear coefficients) using a convolutional layer with a kernel size of 31.

## B.2 Training details

In total, we used 733 cat videos and 753 dog videos for training. We trained our networks for 7000 and 3500 epochs for the single-class and multi-class setups respectively. Training our method lasts about one week on a single Tesla V100 GPU with 32GB memory. We used the Adam optimizer [23] with a learning rate of $10^{-4}$. Our method assumes known camera internal parameters which are provided by the dataset and used to normalize the point tracks as a preprocessing step.

## B.3 Licenses and links for existing assets

**CoTracker[20]** The code is available here: https://github.com/facebookresearch/co-tracker. It is released under Creative Commons Attribution-NonCommercial 4.0 International Public License

**Common pets dataset [44]** The data is available here: https://github.com/facebookresearch/cop3d. It is released under Creative Commons Attribution-NonCommercial 4.0 International Public License.

**Nvidia Dynamic Scenes Dataset [62]** The data is available here:

https://gorokee.github.io/jsyoon/dynamic_synth/. It is released under the CC_BY-NC-ND License.

Table 6: **PointOdyssey**. Average results on 9 test samples from the PointOdyssey dataset [68]. Metrics are the same as in the main paper. Note that since the input point tracks are provided by the dataset, our running times are computed without considering point track extraction time. As can be seen, our model trained only on cats, can efficiently and accurately handle sequences from the PointOdyssey dataset.

| | Abs Rel ↓ | | $\delta < 1.25$ ↑ | | $\delta < 1.25^2$ ↑ | | $\delta < 1.25^3$ ↑ | | ATE ↓ | RPE Trans ↓ | RPE Rot ↓ | Time ↓ |
| | Dyn. | All | Dyn. | All | Dyn. | All | Dyn. | All | | | (deg) | (minutes) |
|---|---|---|---|---|---|---|---|---|---|---|---|---|
| Ours (C) | **0.09** | **0.09** | **0.91** | **0.92** | **1.00** | **1.00** | **1.00** | **1.00** | 0.05 | 0.02 | 0.18 | **0.003** |
| Ours (C) + BA | **0.09** | **0.09** | **0.91** | **0.92** | **1.00** | **1.00** | **1.00** | **1.00** | **0.02** | **0.01** | 0.06 | 0.007 |
| CasualSAM [64] | 0.12 | 0.11 | 0.85 | 0.86 | 0.98 | 0.99 | 0.99 | 1.00 | **0.02** | **0.01** | **0.03** | 62 |

Table 7: **Comparisons with Marigold**. Comparison with the Marigold depth prediction method [21] on the pet test set. As can be seen, our depth accuracy is much higher.

| | Abs Rel ↓ | | $\delta < 1.25$ ↑ | | $\delta < 1.25^2$ ↑ | | $\delta < 1.25^3$ ↑ | |
| | Dyn. | All | Dyn. | All | Dyn. | All | Dyn. | All |
|---|---|---|---|---|---|---|---|---|
| Ours (C) | **0.11** | **0.08** | **0.88** | **0.92** | **0.99** | **0.98** | **1.00** | **1.00** |
| Marigold | 0.48 | 0.31 | 0.28 | 0.51 | 0.50 | 0.75 | 0.64 | 0.86 |

## B.4 Other implementation details

**Point tracks sampling**   For building $M \in \mathbb{R}^{N \times P \times 3}$ we use the implementation of [20]. We sample a uniform grid of $15 \times 15$ 2D points, starting from frame number $0, 20, 40, \dots$, and then track these points throughout the entire video (backward and forward). In Tab. 8 we show the effect of reducing the grid size at inference time, in terms of camera pose accuracy and running time. During training, at each iteration, we randomly sample 20-50 frames from the training videos and 100 point tracks, i.e. $20 \leq N \leq 50$ and $P = 100$. When sampling cameras and point tracks of size $N \times P \times 3$ from a larger tensor of size $N' \times P' \times 3$ we only take a point track if its starting tracking time is in the range $[t - \frac{N}{2}, t + \frac{3N}{2}]$, where t is the first sampled index. At inference time we take all the available point tracks. In both, training and inference time, we keep only point tracks that are observed in more than 10 frames.

**Finetuning details**   For our fine-tuning (FT) in the main paper, we applied per-sequence fine-tunning of 500,100 iterations starting from our final checkpoint, for pets,out-of-domain data respectively. The fine-tuning is done as a post-processing by minimizing the original loss function on the given test video.

**Test Set**   We used the RGBD camera of the iPhone 11 to record our 21 test videos of dogs and cats. Each frame has a resolution of $640 \times 480$ pixels. Note that the training set contained various types of resolutions. All pet owners who were photographed gave their permission for the animals to be photographed. For evaluation only, we define a point track as dynamic if its associated GT mask value is 1 for at least 40 frames. The GT masks are obtained by running [71] and searching for labels of dogs and cats. They were only used for evaluation and not used by our method at all. We verified that this data includes enough dynamic motion, by also including several videos that COLMAP failed to reconstruct without the masks. We further verified manually that the camera trajectories look reasonable.

The out-of-domain dataset contains video sequences with 24 frames, each of resolution of $546 \times 288$. The GT dynamic masks are provided by the dataset. For this dataset, for evaluation only, we define a point track as dynamic if its associated GT mask value is 1 for at least 15 frames.

**Bundle Adjustment**   After inference, as optional refinement, we take the output static approximation $B_1 \in \mathbb{R}^{P \times 3}$ and the output camera poses $\{R_1, \dots, R_N\}$, $\{\mathbf{t}_1, \dots, \mathbf{t}_N\}$ and apply Bundle Adjustment (BA). We use a 3D world point from $B_1$ if its associated $\gamma$ is below 0.008 for the pets dataset and 0.005 for out-of-domain dataset. We optimize reprojection errors of a given observation $M_{i,j}$, only if it is observed, i.e. $M_{i,j}^o = 1$ and if the initial reprojection error is below 10 pixels. We use the BA implementation provided by [32], which is based on the Ceres package [2].

**Running times**   All inference running times were computed on a machine with NVIDIA RTX A6000 GPU and Intel(R) Core(TM) i7-9800X 3.80GHz CPU. Extracting point tracks with [20] took

| | Grid size | ATE ↓ (mm) | RPE Trans ↓ (mm) | RPE Rot ↓ (deg) | Inference ↓ Time |
|---|---|---|---|---|---|
| Ours (C) | 15 | 8.96 | 3.79 | 0.23 | 0.16(+8.6) seconds |
| Ours (C)+BA | 15 | 4.22 | 2.86 | 0.17 | 0.40(+8.6) seconds |
| Ours (C) | 12 | 9.18 | 3.81 | 0.24 | 0.09(+7.8) seconds |
| Ours (C)+BA | 12 | 4.36 | 2.97 | 0.17 | 0.24(+7.8) seconds |
| Ours (C) | 10 | 8.91 | 3.91 | 0.23 | 0.05(+7.7) seconds |
| Ours (C)+BA | 10 | 4.44 | 3.01 | 0.18 | 0.16(+7.7) seconds |
| Ours (C) | 7 | 9.06 | 4.11 | 0.25 | 0.02(+7.6) seconds |
| Ours (C)+BA | 7 | 4.93 | 3.52 | 0.20 | 0.08(+7.6) seconds |
| Ours (C) | 5 | 10.29 | 4.97 | 0.31 | 0.01(+7.6) seconds |
| Ours (C)+BA | 5 | 8.08 | 6.33 | 0.38 | 0.05(+7.6) seconds |

Table 8: **Tracking Grid Size Effect** Quantitative evaluation of the effect of reducing the number of sampled point tracks at inference time on our model that was trained on cats (C). We measure the camera pose accuracy and the running time. We also mention the point tracks extraction time in parenthesis (e.g. +8.6 seconds) which is performed by [20] as a preprocess. As can be seen, our method can handle a smaller number of points but the accuracy slightly drops with fewer sampled points

8.6 and 2.5 seconds for each video on the pet-videos and out-of-domain videos respectively and included in the running time tables as part of our method inference time.

**Training technical details** In all training setups, we used: $\lambda_{\text{Reprojection}} = 50.0$, $\lambda_{\text{Static}} = 1.0$, $\lambda_{\text{Negative}} = 1.0$, $\lambda_{\text{Sparse}} = 0.001$. At the beginning of the training, we pre-train the camera poses to be located behind and facing the origin. This prevents cases in which the cameras are located in the middle of the initial point cloud s.t. many points have negative depths, which may result in bad convergence. More specifically, the pre-train loss is: $\mathcal{L}_{\text{Pretrain}} = \frac{1}{N} \sum_{i=1}^{N} \frac{1}{100} \left\| \mathbf{t}_i - [0,0,-15]^T \right\|^2 + \left\| R_i - I \right\|_F^2$.

The pretrain runs until convergence ($\mathcal{L}_{\text{Pretrain}} < 10^{-4}$). During the main training, we detach gradients from $B_1$ and $(R_1, \mathbf{t}_1) \ldots, (R_N, \mathbf{t}_N)$ for $\mathcal{L}_{\text{Reproject}}$ to stabilize the training. Until epoch 50 we sample sequences of $N$ in the range $[20, 22]$, and then we increase the range to $[20, 50]$.

## C  Broader Impacts

Our work focuses on mapping video data to dynamic 3D point clouds and camera motion. Our network efficiently takes this video input and generates dynamic point cloud representations that model the evolving 3D structure as a set of points in space, enabling analysis of geometry changes across the video frames. This could improve applications in domains where a fast understanding of the spatiotemporal evolution of 3D structure from video data is valuable, such as motion capture, robotic perception, augmented reality, and dynamic scene reconstruction. As robotic capabilities advance, it could fundamentally reshape how society perceives and interacts with the domains that become automated by robots, as well as the individuals previously employed in those domains being replaced.

| | | | RCVD [25] | MiDaS[5] | CasualSAM[64] | Ours (C&D) | Ours (C&D) FT | Our (C) | Our (C) FT |
|---|---|---|---|---|---|---|---|---|---|
| Seq0 | Abs Rel↓ | Dyn | 0.11 | 0.12 | **0.05** | **0.05** | 0.06 | **0.05** | 0.06 |
| | | All | 1.90E+08 | 8.80E+05 | 0.11 | **0.06** | **0.06** | **0.06** | **0.06** |
| | $\delta < 1.25$ ↑ | Dyn | 0.94 | 0.87 | **1.00** | 0.99 | 0.98 | 0.99 | 0.98 |
| | | All | 0.71 | 0.74 | 0.96 | **0.98** | 0.97 | **0.98** | 0.97 |
| | $\delta < 1.25^2$ ↑ | Dyn | **1.00** | **1.00** | **1.00** | **1.00** | **1.00** | **1.00** | **1.00** |
| | | All | 0.85 | 0.87 | 0.97 | **1.00** | 0.99 | **1.00** | 0.99 |
| | $\delta < 1.25^3$ ↑ | Dyn | **1.00** | **1.00** | **1.00** | **1.00** | **1.00** | **1.00** | **1.00** |
| | | All | 0.89 | 0.91 | 0.97 | **1.00** | **1.00** | **1.00** | **1.00** |
| Seq1 | Abs Rel↓ | Dyn | 0.29 | 0.16 | 0.16 | 0.14 | 0.12 | 0.13 | **0.11** |
| | | All | 0.19 | 0.18 | 0.09 | 0.09 | **0.07** | 0.09 | **0.07** |
| | $\delta < 1.25$ ↑ | Dyn | 0.45 | 0.75 | 0.81 | 0.81 | 0.86 | **0.90** | 0.87 |
| | | All | 0.68 | 0.76 | 0.92 | 0.90 | 0.92 | **0.94** | 0.92 |
| | $\delta < 1.25^2$ ↑ | Dyn | 0.85 | 0.99 | 0.89 | **1.00** | **1.00** | 0.99 | **1.00** |
| | | All | 0.93 | 0.95 | 0.96 | **1.00** | **1.00** | **1.00** | **1.00** |
| | $\delta < 1.25^3$ ↑ | Dyn | 0.99 | **1.00** | **1.00** | **1.00** | **1.00** | **1.00** | **1.00** |
| | | All | **1.00** | 0.99 | **1.00** | **1.00** | **1.00** | **1.00** | **1.00** |

| Seq | Metric | | | | | | | | |
|---|---|---|---|---|---|---|---|---|---|
| Seq2 | Abs Rel↓ | Dyn | 0.54 | 0.10 | 0.06 | 0.06 | **0.03** | **0.03** | 0.04 |
| | | All | 0.20 | 0.55 | **0.06** | 0.07 | **0.06** | **0.06** | **0.06** |
| | $\delta < 1.25$ ↑ | Dyn | 0.20 | 0.94 | **0.99** | **0.99** | 0.99 | **0.99** | 0.99 |
| | | All | 0.66 | 0.66 | **0.98** | 0.96 | **0.98** | 0.97 | **0.98** |
| | $\delta < 1.25^2$ ↑ | Dyn | 0.52 | **1.00** | 0.99 | **1.00** | **1.00** | **1.00** | **1.00** |
| | | All | 0.89 | 0.75 | **0.99** | 0.99 | 0.99 | 0.99 | 0.99 |
| | $\delta < 1.25^3$ ↑ | Dyn | 0.91 | **1.00** | 1.00 | 1.00 | 1.00 | 1.00 | 1.00 |
| | | All | 0.98 | 0.80 | **0.99** | 0.99 | 0.99 | 0.99 | 0.99 |
| Seq3 | Abs Rel↓ | Dyn | 0.79 | 0.25 | 0.07 | 0.15 | **0.05** | 0.07 | **0.05** |
| | | All | 0.22 | 0.24 | **0.06** | 0.09 | **0.06** | 0.08 | **0.06** |
| | $\delta < 1.25$ ↑ | Dyn | 0.10 | 0.53 | 0.97 | 0.95 | **0.99** | 0.97 | 0.98 |
| | | All | 0.76 | 0.74 | **0.97** | 0.91 | **0.97** | 0.92 | **0.97** |
| | $\delta < 1.25^2$ ↑ | Dyn | 0.27 | 0.99 | **1.00** | **1.00** | **1.00** | **1.00** | **1.00** |
| | | All | 0.85 | 0.92 | **0.99** | 0.98 | **0.99** | 0.98 | **0.99** |
| | $\delta < 1.25^3$ ↑ | Dyn | 0.59 | **1.00** | 1.00 | 1.00 | 1.00 | 1.00 | 1.00 |
| | | All | 0.92 | 0.95 | **0.99** | 0.99 | 0.99 | 0.99 | 0.99 |
| Seq4 | Abs Rel↓ | Dyn | 0.31 | 0.08 | **0.06** | 0.27 | 0.15 | 0.19 | 0.15 |
| | | All | 0.17 | 0.27 | **0.09** | 0.12 | 0.11 | 0.10 | 0.11 |
| | $\delta < 1.25$ ↑ | Dyn | 0.44 | 0.98 | **1.00** | 0.54 | 0.75 | 0.65 | 0.77 |
| | | All | 0.80 | 0.65 | **0.96** | 0.84 | 0.92 | 0.89 | 0.92 |
| | $\delta < 1.25^2$ ↑ | Dyn | 0.87 | **1.00** | **1.00** | 0.88 | **1.00** | 0.99 | **1.00** |
| | | All | 0.97 | 0.90 | 0.98 | 0.98 | 0.98 | **1.00** | 0.99 |
| | $\delta < 1.25^3$ ↑ | Dyn | **1.00** | **1.00** | **1.00** | **1.00** | **1.00** | **1.00** | **1.00** |
| | | All | **1.00** | 0.94 | 0.99 | **1.00** | **1.00** | **1.00** | **1.00** |
| Seq5 | Abs Rel↓ | Dyn | 0.09 | 0.08 | **0.05** | 0.07 | 0.07 | 0.06 | 0.07 |
| | | All | 0.12 | 3.75E+05 | **0.03** | 0.08 | 0.05 | 0.06 | 0.04 |
| | $\delta < 1.25$ ↑ | Dyn | 0.97 | 0.98 | **1.00** | 0.99 | 0.98 | 0.99 | 0.98 |
| | | All | 0.86 | 0.86 | **0.98** | 0.91 | 0.97 | 0.96 | **0.98** |
| | $\delta < 1.25^2$ ↑ | Dyn | **1.00** | **1.00** | **1.00** | **1.00** | **1.00** | **1.00** | **1.00** |
| | | All | 0.97 | 0.95 | **1.00** | 0.97 | **1.00** | 0.98 | **1.00** |
| | $\delta < 1.25^3$ ↑ | Dyn | **1.00** | **1.00** | **1.00** | **1.00** | **1.00** | **1.00** | **1.00** |
| | | All | 0.99 | 0.98 | **1.00** | 0.99 | **1.00** | **1.00** | **1.00** |
| Seq6 | Abs Rel↓ | Dyn | 0.35 | 0.10 | 0.04 | 0.05 | **0.03** | 0.04 | **0.03** |
| | | All | 0.14 | 0.15 | 0.05 | **0.04** | 0.05 | 0.05 | 0.05 |
| | $\delta < 1.25$ ↑ | Dyn | 0.47 | 0.94 | 0.98 | **0.99** | **0.99** | **0.99** | **0.99** |
| | | All | 0.83 | 0.87 | 0.96 | **0.98** | 0.96 | 0.97 | 0.96 |
| | $\delta < 1.25^2$ ↑ | Dyn | 0.65 | 0.99 | 0.99 | **1.00** | **1.00** | **1.00** | **1.00** |
| | | All | 0.91 | 0.95 | **0.99** | **0.99** | 0.98 | **0.99** | 0.98 |
| | $\delta < 1.25^3$ ↑ | Dyn | 0.99 | 0.99 | **1.00** | **1.00** | **1.00** | **1.00** | **1.00** |
| | | All | 0.99 | 0.98 | **1.00** | **1.00** | 0.99 | **1.00** | 0.99 |
| Seq7 | Abs Rel↓ | Dyn | 0.39 | **0.08** | 0.09 | 0.15 | 0.11 | 0.12 | 0.10 |
| | | All | 0.22 | 0.17 | **0.06** | 0.10 | **0.06** | 0.09 | **0.06** |
| | $\delta < 1.25$ ↑ | Dyn | 0.45 | **0.95** | 0.92 | 0.84 | 0.91 | 0.90 | 0.91 |
| | | All | 0.65 | 0.80 | **0.97** | 0.86 | 0.96 | 0.91 | **0.97** |
| | $\delta < 1.25^2$ ↑ | Dyn | 0.68 | **1.00** | 0.99 | 0.98 | 0.98 | 0.98 | 0.98 |
| | | All | 0.86 | 0.92 | **0.99** | 0.98 | **0.99** | 0.98 | **0.99** |
| | $\delta < 1.25^3$ ↑ | Dyn | 0.93 | **1.00** | **1.00** | **1.00** | 0.99 | **1.00** | 0.99 |
| | | All | 0.97 | 0.98 | **1.00** | 0.99 | **1.00** | 0.99 | **1.00** |
| Seq8 | Abs Rel↓ | Dyn | 0.47 | 0.23 | **0.05** | 0.09 | **0.05** | 0.09 | **0.05** |
| | | All | 0.20 | 3.80E+04 | **0.03** | 0.08 | 0.04 | 0.07 | 0.04 |
| | $\delta < 1.25$ ↑ | Dyn | 0.18 | 0.68 | **0.99** | 0.97 | **0.99** | 0.98 | **0.99** |
| | | All | 0.72 | 0.64 | **0.99** | 0.89 | **0.99** | 0.97 | **0.99** |
| | $\delta < 1.25^2$ ↑ | Dyn | 0.69 | 0.97 | **1.00** | **1.00** | **1.00** | **1.00** | **1.00** |
| | | All | 0.91 | 0.84 | **1.00** | 0.99 | **1.00** | **1.00** | **1.00** |
| | $\delta < 1.25^3$ ↑ | Dyn | 0.97 | **1.00** | **1.00** | **1.00** | **1.00** | **1.00** | **1.00** |
| | | All | 0.99 | 0.92 | **1.00** | **1.00** | **1.00** | **1.00** | **1.00** |
| Seq9 | Abs Rel↓ | Dyn | 0.86 | 0.26 | 0.18 | 0.22 | **0.17** | 0.22 | 0.18 |
| | | All | 0.33 | 0.20 | **0.09** | 0.21 | **0.09** | 0.22 | **0.09** |
| | $\delta < 1.25$ ↑ | Dyn | 0.05 | 0.59 | **0.80** | 0.46 | 0.63 | 0.45 | 0.60 |
| | | All | 0.65 | 0.74 | **0.92** | 0.55 | 0.88 | 0.52 | 0.87 |
| | $\delta < 1.25^2$ ↑ | Dyn | 0.28 | 0.85 | 0.95 | 0.91 | **1.00** | 0.85 | **1.00** |
| | | All | 0.78 | 0.91 | 0.98 | 0.89 | **0.99** | 0.86 | 0.98 |
| | $\delta < 1.25^3$ ↑ | Dyn | 0.68 | **1.00** | **1.00** | **1.00** | **1.00** | **1.00** | **1.00** |
| | | All | 0.90 | 0.98 | **1.00** | 0.97 | **1.00** | 0.97 | 0.99 |
| Seq10 | Abs Rel↓ | Dyn | 0.08 | 0.06 | **0.02** | **0.02** | 0.02 | 0.02 | 0.02 |
| | | All | 0.11 | 0.24 | 0.03 | 0.03 | **0.02** | 0.04 | **0.02** |
| | $\delta < 1.25$ ↑ | Dyn | 0.99 | **1.00** | **1.00** | **1.00** | **1.00** | **1.00** | **1.00** |
| | | All | 0.92 | 0.73 | **1.00** | **1.00** | **1.00** | 0.99 | **1.00** |
| | $\delta < 1.25^2$ ↑ | Dyn | **1.00** | **1.00** | **1.00** | **1.00** | **1.00** | **1.00** | **1.00** |
| | | All | 0.99 | 0.87 | **1.00** | **1.00** | **1.00** | **1.00** | **1.00** |
| | $\delta < 1.25^3$ ↑ | Dyn | **1.00** | **1.00** | **1.00** | **1.00** | **1.00** | **1.00** | **1.00** |
| | | All | **1.00** | 0.95 | **1.00** | **1.00** | **1.00** | **1.00** | **1.00** |
| Seq11 | Abs Rel↓ | Dyn | 0.34 | 0.14 | **0.06** | 0.10 | 0.07 | 0.10 | 0.07 |
| | | All | 0.17 | 0.38 | **0.05** | 0.08 | **0.05** | 0.07 | **0.05** |
| | $\delta < 1.25$ ↑ | Dyn | 0.47 | 0.79 | **0.92** | 0.90 | **0.92** | 0.90 | **0.92** |
| | | All | 0.74 | 0.67 | **0.97** | 0.90 | 0.96 | 0.94 | 0.96 |
| | $\delta < 1.25^2$ ↑ | Dyn | 0.86 | **1.00** | 0.99 | 0.96 | 0.98 | 0.97 | 0.98 |
| | | All | 0.94 | 0.78 | **0.99** | 0.99 | **0.99** | **0.99** | **0.99** |
| | $\delta < 1.25^3$ ↑ | Dyn | 0.94 | **1.00** | **1.00** | 0.99 | **1.00** | 0.99 | **1.00** |
| | | All | 0.99 | 0.86 | **1.00** | **1.00** | **1.00** | **1.00** | **1.00** |
| Seq12 | Abs Rel↓ | Dyn | 0.37 | 0.16 | 0.05 | 0.10 | 0.05 | 0.09 | **0.04** |
| | | All | 0.16 | 0.13 | **0.05** | 0.08 | 0.06 | 0.08 | 0.07 |
| | $\delta < 1.25$ ↑ | Dyn | 0.19 | 0.79 | **0.98** | **0.98** | 0.97 | **0.98** | 0.97 |
| | | All | 0.73 | 0.85 | **0.98** | 0.95 | 0.94 | 0.95 | 0.94 |
| | $\delta < 1.25^2$ ↑ | Dyn | 0.97 | **1.00** | **1.00** | **1.00** | **1.00** | **1.00** | **1.00** |
| | | All | 0.98 | 0.98 | **1.00** | 0.99 | 0.98 | 0.99 | 0.98 |
| | $\delta < 1.25^3$ ↑ | Dyn | **1.00** | **1.00** | **1.00** | **1.00** | **1.00** | **1.00** | **1.00** |
| | | All | **1.00** | **1.00** | **1.00** | **1.00** | **1.00** | **1.00** | **1.00** |
| Seq13 | Abs Rel↓ | Dyn | 0.31 | 0.29 | **0.09** | 0.15 | 0.11 | 0.14 | 0.11 |
| | | All | 3.31E+08 | 0.35 | **0.06** | 0.10 | 0.07 | 0.09 | 0.07 |
| | $\delta < 1.25$ ↑ | Dyn | 0.50 | 0.50 | 0.93 | 0.82 | 0.92 | 0.85 | **0.95** |

| Seq | Metric | | | | | | | | |
|---|---|---|---|---|---|---|---|---|---|
| | | All | 0.67 | 0.69 | **0.97** | 0.91 | 0.96 | 0.93 | **0.97** |
| | $\delta < 1.25^2 \uparrow$ | Dyn | 0.81 | 0.84 | **1.00** | **1.00** | **1.00** | **1.00** | **1.00** |
| | | All | 0.81 | 0.87 | **0.99** | 0.98 | **0.99** | **0.99** | **0.99** |
| | $\delta < 1.25^3 \uparrow$ | Dyn | **1.00** | **1.00** | **1.00** | **1.00** | **1.00** | **1.00** | **1.00** |
| | | All | 0.87 | 0.93 | **1.00** | 0.99 | 0.99 | 0.99 | 0.99 |
| Seq14 | Abs Rel↓ | Dyn | 0.18 | 0.17 | **0.03** | 0.05 | 0.04 | 0.04 | 0.04 |
| | | All | 0.18 | 0.34 | **0.03** | 0.04 | **0.03** | **0.03** | **0.03** |
| | $\delta < 1.25 \uparrow$ | Dyn | 0.79 | 0.72 | **0.99** | **0.99** | **0.99** | **0.99** | **0.99** |
| | | All | 0.76 | 0.55 | **1.00** | 0.99 | 0.99 | 0.99 | 0.99 |
| | $\delta < 1.25^2 \uparrow$ | Dyn | 0.93 | 0.97 | **1.00** | **1.00** | **1.00** | **1.00** | **1.00** |
| | | All | 0.94 | 0.83 | **1.00** | **1.00** | **1.00** | **1.00** | **1.00** |
| | $\delta < 1.25^3 \uparrow$ | Dyn | 0.99 | **1.00** | **1.00** | **1.00** | **1.00** | **1.00** | **1.00** |
| | | All | 0.99 | 0.92 | **1.00** | **1.00** | **1.00** | **1.00** | **1.00** |
| Seq15 | Abs Rel↓ | Dyn | 1.22 | 0.12 | 0.15 | 0.27 | **0.10** | 0.18 | **0.10** |
| | | All | 0.33 | 0.39 | **0.09** | 0.15 | **0.09** | 0.18 | **0.09** |
| | $\delta < 1.25 \uparrow$ | Dyn | 0.01 | 0.77 | 0.95 | 0.62 | 0.96 | 0.79 | **0.97** |
| | | All | 0.65 | 0.65 | **0.97** | 0.80 | 0.94 | 0.70 | 0.95 |
| | $\delta < 1.25^2 \uparrow$ | Dyn | 0.12 | **1.00** | 0.99 | 0.93 | 0.99 | 0.96 | 0.99 |
| | | All | 0.81 | 0.83 | **0.99** | 0.97 | **0.99** | 0.92 | **0.99** |
| | $\delta < 1.25^3 \uparrow$ | Dyn | 0.46 | **1.00** | 0.99 | **1.00** | **1.00** | **1.00** | **1.00** |
| | | All | 0.90 | 0.89 | **0.99** | **0.99** | **0.99** | **0.99** | **0.99** |
| Seq16 | Abs Rel↓ | Dyn | 0.28 | 0.12 | **0.07** | 0.14 | 0.11 | 0.14 | 0.11 |
| | | All | 0.21 | 0.21 | **0.05** | 0.06 | 0.06 | 0.06 | 0.06 |
| | $\delta < 1.25 \uparrow$ | Dyn | 0.42 | 0.92 | **0.98** | 0.81 | 0.88 | 0.80 | 0.90 |
| | | All | 0.66 | 0.79 | **0.99** | 0.96 | 0.97 | 0.96 | 0.97 |
| | $\delta < 1.25^2 \uparrow$ | Dyn | 0.98 | **1.00** | **1.00** | **1.00** | **1.00** | **1.00** | **1.00** |
| | | All | 0.95 | 0.93 | **1.00** | **1.00** | **1.00** | **1.00** | **1.00** |
| | $\delta < 1.25^3 \uparrow$ | Dyn | **1.00** | **1.00** | **1.00** | **1.00** | **1.00** | **1.00** | **1.00** |
| | | All | **1.00** | 0.96 | **1.00** | **1.00** | **1.00** | **1.00** | **1.00** |
| Seq17 | Abs Rel↓ | Dyn | 0.35 | 0.12 | 0.14 | **0.11** | 0.12 | **0.11** | 0.12 |
| | | All | 0.23 | 0.15 | 0.08 | 0.07 | **0.06** | 0.07 | 0.07 |
| | $\delta < 1.25 \uparrow$ | Dyn | 0.40 | 0.88 | 0.76 | **0.93** | 0.89 | 0.91 | 0.88 |
| | | All | 0.60 | 0.81 | 0.89 | **0.94** | **0.94** | 0.93 | **0.94** |
| | $\delta < 1.25^2 \uparrow$ | Dyn | 0.82 | **0.99** | 0.98 | **0.99** | 0.98 | 0.98 | 0.98 |
| | | All | 0.87 | 0.94 | 0.98 | 0.98 | **0.99** | **0.99** | **0.99** |
| | $\delta < 1.25^3 \uparrow$ | Dyn | 0.95 | **1.00** | **1.00** | 0.99 | 0.99 | 0.99 | 0.99 |
| | | All | 0.95 | 0.98 | **1.00** | 0.99 | **1.00** | **1.00** | **1.00** |
| Seq18 | Abs Rel↓ | Dyn | 0.48 | 0.21 | 0.10 | 0.12 | **0.07** | 0.11 | **0.07** |
| | | All | 0.14 | 0.32 | **0.05** | 0.10 | 0.09 | 0.08 | 0.09 |
| | $\delta < 1.25 \uparrow$ | Dyn | 0.46 | 0.60 | 0.93 | 0.87 | **0.96** | 0.88 | **0.96** |
| | | All | 0.86 | 0.65 | **0.98** | 0.91 | 0.96 | 0.95 | 0.96 |
| | $\delta < 1.25^2 \uparrow$ | Dyn | 0.55 | 0.95 | 0.98 | **1.00** | 0.99 | 0.99 | 0.99 |
| | | All | 0.91 | 0.84 | **0.99** | 0.98 | 0.98 | 0.98 | 0.98 |
| | $\delta < 1.25^3 \uparrow$ | Dyn | 0.84 | **1.00** | 0.99 | **1.00** | **1.00** | **1.00** | **1.00** |
| | | All | 0.97 | 0.92 | **1.00** | 0.99 | 0.98 | 0.99 | 0.98 |
| Seq19 | Abs Rel↓ | Dyn | 0.33 | 0.14 | 0.06 | **0.04** | 0.05 | 0.05 | 0.05 |
| | | All | 0.15 | 0.27 | **0.03** | **0.03** | **0.03** | 0.05 | **0.03** |
| | $\delta < 1.25 \uparrow$ | Dyn | 0.28 | 0.80 | 0.97 | 0.97 | 0.97 | **0.98** | 0.97 |
| | | All | 0.81 | 0.74 | **0.99** | **0.99** | **0.99** | **0.99** | **0.99** |
| | $\delta < 1.25^2 \uparrow$ | Dyn | 0.91 | 0.97 | **1.00** | **1.00** | 0.99 | 0.99 | 0.99 |
| | | All | 0.98 | 0.87 | **1.00** | **1.00** | **1.00** | **1.00** | **1.00** |
| | $\delta < 1.25^3 \uparrow$ | Dyn | **1.00** | 0.99 | **1.00** | **1.00** | **1.00** | **1.00** | **1.00** |
| | | All | **1.00** | 0.92 | **1.00** | **1.00** | **1.00** | **1.00** | **1.00** |
| Seq20 | Abs Rel↓ | Dyn | 0.33 | 0.27 | **0.21** | 0.27 | 0.29 | 0.26 | 0.30 |
| | | All | 2.42E+08 | 0.45 | **0.04** | 0.05 | 0.05 | 0.05 | 0.05 |
| | $\delta < 1.25 \uparrow$ | Dyn | 0.34 | 0.49 | **0.61** | 0.44 | 0.37 | 0.49 | 0.34 |
| | | All | 0.50 | 0.42 | **0.97** | 0.95 | 0.95 | 0.95 | 0.95 |
| | $\delta < 1.25^2 \uparrow$ | Dyn | 0.93 | 0.90 | **1.00** | **1.00** | **1.00** | **1.00** | **1.00** |
| | | All | 0.77 | 0.72 | **1.00** | **1.00** | **1.00** | **1.00** | **1.00** |
| | $\delta < 1.25^3 \uparrow$ | Dyn | 0.99 | 0.99 | **1.00** | **1.00** | **1.00** | **1.00** | **1.00** |
| | | All | 0.83 | 0.85 | **1.00** | **1.00** | **1.00** | **1.00** | **1.00** |
| Mean | Abs Rel↓ | Dyn | 0.40 | 0.16 | **0.09** | 0.12 | **0.09** | 0.11 | **0.09** |
| | | All | 3.63E+07 | 6.16E+04 | **0.06** | 0.08 | **0.06** | 0.08 | **0.06** |
| | $\delta < 1.25 \uparrow$ | Dyn | 0.43 | 0.78 | **0.93** | 0.85 | 0.90 | 0.88 | 0.90 |
| | | All | 0.72 | 0.71 | **0.97** | 0.91 | 0.96 | 0.92 | 0.96 |
| | $\delta < 1.25^2 \uparrow$ | Dyn | 0.75 | 0.97 | 0.99 | 0.98 | **1.00** | 0.99 | **1.00** |
| | | All | 0.90 | 0.88 | **0.99** | 0.98 | **0.99** | 0.98 | **0.99** |
| | $\delta < 1.25^3 \uparrow$ | Dyn | 0.92 | **1.00** | **1.00** | **1.00** | **1.00** | **1.00** | **1.00** |
| | | All | 0.96 | 0.93 | **1.00** | **1.00** | **1.00** | **1.00** | **1.00** |
| STD | Abs Rel↓ | Dyn | 0.27 | 0.07 | **0.05** | 0.08 | 0.06 | 0.07 | 0.06 |
| | | All | 9.39E+07 | 2.04E+05 | **0.02** | 0.04 | **0.02** | 0.04 | **0.02** |
| | $\delta < 1.25 \uparrow$ | Dyn | 0.29 | 0.17 | **0.10** | 0.18 | 0.15 | 0.16 | 0.16 |
| | | All | 0.10 | 0.11 | **0.03** | 0.10 | **0.03** | 0.11 | **0.03** |
| | $\delta < 1.25^2 \uparrow$ | Dyn | 0.26 | 0.05 | 0.02 | 0.03 | **0.01** | 0.03 | **0.01** |
| | | All | 0.07 | 0.07 | **0.01** | 0.02 | **0.01** | 0.03 | **0.01** |
| | $\delta < 1.25^3 \uparrow$ | Dyn | 0.15 | **0.00** | **0.00** | **0.00** | **0.00** | **0.00** | **0.00** |
| | | All | 0.05 | 0.05 | 0.01 | 0.01 | **0.00** | 0.01 | **0.00** |

Table 9: **Depth accuracy, pet test set.** We show a comparison to previous methods on the predicted depth for the point trajectories compared to their GT depths. We compare 4 ways of running our method. Ours (C&D), Ours (C): Our inference time outputs for a model that was trained on cats and dogs or only on cats respectively. Ours (C&D) FT, Ours (C) FT: The outputs of our model trained on cats and dogs or cats respectively, after fine-tuning our losses for each specific video. As can be seen, fine-tuning improves our accuracy even more.

| | | DROID-SLAM[48] | ParticleSfM[66] | RCVD[25] | CasualSAM[64] | Ours (C) | Ours (C)+BA | Ours (C)+FT |
|---|---|---|---|---|---|---|---|---|
| Seq0 | ATE(mm) | **3.71** | 6.10 | 64.67 | 5.36 | 5.60 | 5.43 | 4.42 |
| | RPE T.(mm) | 3.05 | 3.22 | 26.92 | 3.13 | 3.63 | 3.04 | **2.89** |
| | RPE R.(deg.) | **0.14** | 0.18 | 2.53 | 0.16 | 0.22 | 0.15 | **0.14** |
| Seq1 | ATE(mm) | 1.91 | 3.83 | 38.44 | 10.28 | 13.21 | 4.32 | **1.76** |
| | RPE T.(mm) | **1.32** | 1.49 | 25.23 | 3.00 | 2.65 | 1.72 | **1.32** |
| | RPE R.(deg.) | 0.18 | 0.23 | 2.43 | 0.67 | 0.29 | 0.19 | **0.16** |
| Seq2 | ATE(mm) | 3.13 | 4.68 | 39.46 | **2.13** | 4.57 | 2.78 | 2.53 |
| | RPE T.(mm) | 3.62 | 5.82 | 27.27 | **1.97** | 3.24 | 2.62 | 2.49 |
| | RPE R.(deg.) | 0.08 | 0.21 | 1.89 | **0.06** | 0.12 | 0.09 | 0.09 |
| Seq3 | ATE(mm) | 5.13 | **2.26** | 29.55 | 2.49 | 7.83 | 2.66 | 2.92 |
| | RPE T.(mm) | 5.76 | **2.00** | 24.18 | 2.31 | 3.13 | 2.27 | 2.30 |
| | RPE R.(deg.) | 0.17 | **0.07** | 2.38 | 0.08 | 0.13 | **0.07** | **0.07** |
| Seq4 | ATE(mm) | 2.59 | 4.38 | 56.16 | 2.65 | 4.21 | 2.38 | **2.28** |
| | RPE T.(mm) | 2.05 | 2.07 | 21.36 | **1.74** | 2.40 | 2.05 | 2.04 |
| | RPE R.(deg.) | 0.11 | 0.11 | 1.05 | **0.09** | 0.15 | 0.11 | 0.11 |
| Seq5 | ATE(mm) | 1.07 | 0.83 | 14.22 | **0.53** | 1.44 | 0.79 | 0.75 |
| | RPE T.(mm) | 1.02 | 0.74 | 6.51 | **0.52** | 0.88 | 0.68 | 0.68 |
| | RPE R.(deg.) | 0.09 | 0.07 | 1.63 | **0.05** | 0.12 | 0.07 | 0.06 |
| Seq6 | ATE(mm) | **26.08** | 31.07 | 48.31 | 26.12 | 27.10 | 28.28 | 28.98 |
| | RPE T.(mm) | 10.57 | 11.01 | 24.21 | **10.31** | 10.44 | 10.92 | 10.82 |
| | RPE R.(deg.) | 0.66 | 0.67 | 4.03 | **0.62** | 0.69 | 0.69 | 0.68 |
| Seq7 | ATE(mm) | 2.25 | 28.48 | 47.25 | **1.78** | 4.53 | 2.39 | 2.25 |
| | RPE T.(mm) | 2.34 | 6.13 | 23.81 | **1.72** | 2.32 | 1.98 | 1.99 |
| | RPE R.(deg.) | 0.15 | 0.59 | 2.29 | **0.11** | 0.20 | 0.16 | 0.16 |
| Seq8 | ATE(mm) | 1.23 | 38.79 | 44.06 | 2.00 | 4.78 | **0.84** | 0.89 |
| | RPE T.(mm) | 1.07 | 50.57 | 25.46 | 1.24 | 1.54 | **0.71** | 0.70 |
| | RPE R.(deg.) | 0.07 | 4.78 | 1.73 | 0.09 | 0.13 | 0.06 | **0.05** |
| Seq9 | ATE(mm) | 21.74 | 18.52 | 43.45 | 34.93 | 24.15 | 3.92 | **3.22** |
| | RPE T.(mm) | 9.04 | 7.96 | 21.35 | 21.06 | 6.89 | 2.77 | **2.08** |
| | RPE R.(deg.) | 0.62 | 0.47 | 3.47 | 0.71 | 0.37 | 0.12 | **0.10** |
| Seq10 | ATE(mm) | 1.47 | 1.75 | 22.49 | **1.40** | 3.03 | 1.42 | 1.46 |
| | RPE T.(mm) | 1.84 | 1.15 | 24.22 | **1.11** | 1.87 | 1.25 | 1.22 |
| | RPE R.(deg.) | 0.22 | 0.12 | 2.18 | **0.10** | 0.21 | 0.12 | 0.12 |
| Seq11 | ATE(mm) | 1.71 | 3.60 | 19.10 | **1.32** | 3.12 | 1.66 | 1.49 |
| | RPE T.(mm) | 1.65 | 1.54 | 16.34 | **1.21** | 3.12 | 1.75 | 1.55 |
| | RPE R.(deg.) | 0.08 | 0.08 | 1.80 | **0.06** | 0.12 | 0.08 | 0.07 |
| Seq12 | ATE(mm) | **1.70** | 3.64 | 20.82 | 2.51 | 6.23 | 3.61 | 2.54 |
| | RPE T.(mm) | **2.24** | 2.63 | 18.50 | 2.74 | 4.04 | 3.02 | 2.84 |
| | RPE R.(deg.) | **0.09** | 0.12 | 2.03 | 0.12 | 0.23 | 0.15 | 0.14 |
| Seq13 | ATE(mm) | 1.23 | 2.38 | 33.49 | 1.49 | 4.10 | **1.17** | 1.28 |
| | RPE T.(mm) | 1.19 | 1.40 | 17.33 | **1.00** | 1.72 | 1.12 | 1.09 |
| | RPE R.(deg.) | 0.13 | 0.14 | 2.12 | **0.12** | 0.19 | **0.12** | **0.12** |
| Seq14 | ATE(mm) | 5.42 | 5.15 | 1.05E+02 | 24.95 | 6.09 | 5.06 | **4.93** |
| | RPE T.(mm) | 3.40 | **3.38** | 69.36 | 9.45 | 4.07 | 3.57 | 3.41 |
| | RPE R.(deg.) | **0.18** | 0.19 | 3.81 | 0.65 | 0.26 | 0.20 | 0.19 |
| Seq15 | ATE(mm) | 7.69 | 61.06 | 36.70 | 7.40 | 36.51 | **4.98** | 5.41 |
| | RPE T.(mm) | 7.95 | 17.57 | 28.18 | 6.93 | 10.19 | 5.86 | **5.72** |
| | RPE R.(deg.) | 0.22 | 0.58 | 2.85 | 0.19 | 0.30 | 0.16 | **0.15** |
| Seq16 | ATE(mm) | 5.04 | 5.06 | 36.42 | **3.69** | 4.52 | 4.11 | 3.81 |
| | RPE T.(mm) | 4.53 | 4.54 | 20.11 | **4.02** | 4.47 | 4.25 | 4.15 |
| | RPE R.(deg.) | 0.28 | 0.29 | 2.94 | **0.25** | 0.31 | 0.28 | 0.27 |
| Seq17 | ATE(mm) | **1.12** | 34.07 | 77.91 | 2.08 | 6.52 | 2.51 | 2.86 |
| | RPE T.(mm) | **1.15** | 12.30 | 39.64 | 1.18 | 2.14 | 1.50 | 1.55 |
| | RPE R.(deg.) | 0.10 | 1.11 | 2.20 | **0.08** | 0.17 | 0.13 | 0.13 |
| Seq18 | ATE(mm) | **2.98** | 6.25 | 36.54 | 4.91 | 7.73 | 4.13 | 3.86 |
| | RPE T.(mm) | 4.05 | 5.21 | 24.78 | 3.89 | 4.34 | **3.62** | 3.65 |
| | RPE R.(deg.) | 0.23 | 0.29 | 1.67 | **0.21** | 0.24 | **0.21** | **0.21** |
| Seq19 | ATE(mm) | **1.45** | 2.23 | 42.95 | 1.82 | 8.51 | 2.79 | 2.36 |
| | RPE T.(mm) | 1.65 | 1.72 | 29.12 | **1.44** | 3.23 | 2.21 | 2.01 |
| | RPE R.(deg.) | 0.11 | 0.11 | 2.09 | **0.09** | 0.25 | 0.16 | 0.15 |
| Seq20 | ATE(mm) | 8.13 | 4.37 | 66.03 | 4.95 | 4.30 | **3.43** | 4.06 |
| | RPE T.(mm) | 6.20 | 3.57 | 27.34 | **3.00** | 3.30 | 3.05 | 3.06 |
| | RPE R.(deg.) | 0.36 | 0.20 | 1.35 | **0.18** | 0.20 | **0.18** | **0.18** |
| Mean | ATE(mm) | 5.08 | 12.79 | 43.95 | 6.90 | 8.96 | 4.22 | **4.00** |
| | RPE T.(mm) | 3.60 | 6.95 | 25.77 | 3.95 | 3.79 | 2.86 | **2.74** |
| | RPE R.(deg.) | 0.20 | 0.51 | 2.31 | 0.22 | 0.23 | 0.17 | **0.16** |
| STD | ATE(mm) | 6.63 | 16.32 | 21.22 | 9.55 | 9.06 | **5.68** | 5.87 |
| | RPE T.(mm) | 2.80 | 10.88 | 11.80 | 4.74 | 2.52 | **2.22** | **2.22** |
| | RPE R.(deg.) | 0.16 | 1.01 | 0.76 | 0.22 | **0.13** | **0.13** | **0.13** |

Table 10: **Camera poses accuracy for pets.** We show a comparison to previous methods on the predicted camera poses. We compare 3 ways of running our method. Ours (C): Our inference time outputs (total inference time of 0.16 seconds) for a model that was trained only on cats. Ours (C)+BA: Our inference time outputs, followed by a short Bundle Adjustment (total inference time of 0.4 seconds) for a model that was trained only on cats. Ours (C) FT: The outputs of the model that was trained only on cats after fine-tuning our losses for each specific video (total running time of about 5 minutes). As can be seen, after BA, our results are the most accurate compared to the other methods, and fine-tuning improves our accuracy even more.

| | | | RCVD [25] | MiDaS[5] | CasualSAM[64] | Ours (C&D) | Ours (C&D) FT |
|---|---|---|---|---|---|---|---|
| Balloon1 | Abs Rel↓ | Dyn | 0.21 | 0.12 | 0.04 | 0.09 | **0.03** |
| | | All | 0.14 | 0.34 | 0.02 | 0.06 | **0.01** |
| | $\delta < 1.25$ ↑ | Dyn | 0.42 | 0.89 | **0.98** | **0.98** | **0.98** |
| | | All | 0.62 | 0.72 | **0.99** | **0.99** | **0.99** |
| | $\delta < 1.25^2$ ↑ | Dyn | **1.00** | 0.98 | 0.99 | 0.99 | 0.99 |

| | | | | | | |
|---|---|---|---|---|---|---|
| | | All | **1.00** | 0.81 | **1.00** | **1.00** | **1.00** |
| | $\delta < 1.25^3$ ↑ | Dyn | **1.00** | 0.99 | **1.00** | **1.00** | **1.00** |
| | | All | **1.00** | 0.88 | **1.00** | **1.00** | **1.00** |
| Balloon2 | Abs Rel↓ | Dyn | 0.10 | 2.21E+05 | 0.04 | 0.04 | **0.03** |
| | | All | 0.14 | 4.87E+05 | **0.01** | 0.06 | **0.01** |
| | $\delta < 1.25$ ↑ | Dyn | 0.97 | 0.95 | **1.00** | 0.99 | **1.00** |
| | | All | 0.83 | 0.76 | **1.00** | 0.97 | **1.00** |
| | $\delta < 1.25^2$ ↑ | Dyn | **1.00** | 0.99 | **1.00** | **1.00** | **1.00** |
| | | All | **1.00** | 0.86 | **1.00** | **1.00** | **1.00** |
| | $\delta < 1.25^3$ ↑ | Dyn | **1.00** | 0.99 | **1.00** | **1.00** | **1.00** |
| | | All | **1.00** | 0.93 | **1.00** | **1.00** | **1.00** |
| DynamicFace | Abs Rel↓ | Dyn | 0.14 | 0.55 | **0.01** | 0.15 | **0.01** |
| | | All | 0.05 | 4.75E+04 | **0.01** | 0.06 | **0.01** |
| | $\delta < 1.25$ ↑ | Dyn | 0.94 | 0.03 | **0.99** | 0.98 | 0.98 |
| | | All | 0.98 | 0.67 | **1.00** | **1.00** | **1.00** |
| | $\delta < 1.25^2$ ↑ | Dyn | **1.00** | 0.04 | **1.00** | 0.98 | **1.00** |
| | | All | **1.00** | 0.82 | **1.00** | **1.00** | **1.00** |
| | $\delta < 1.25^3$ ↑ | Dyn | **1.00** | 0.18 | **1.00** | **1.00** | **1.00** |
| | | All | **1.00** | 0.86 | **1.00** | **1.00** | **1.00** |
| Jumping | Abs Rel↓ | Dyn | 0.17 | 0.43 | **0.05** | 0.07 | 0.07 |
| | | All | 0.12 | 0.59 | **0.02** | 0.05 | 0.04 |
| | $\delta < 1.25$ ↑ | Dyn | 0.77 | 0.07 | **0.99** | 0.96 | 0.96 |
| | | All | 0.86 | 0.22 | **0.99** | 0.97 | 0.97 |
| | $\delta < 1.25^2$ ↑ | Dyn | **1.00** | 0.30 | **1.00** | **1.00** | 0.99 |
| | | All | **1.00** | 0.38 | **1.00** | **1.00** | **1.00** |
| | $\delta < 1.25^3$ ↑ | Dyn | **1.00** | 0.75 | **1.00** | **1.00** | **1.00** |
| | | All | **1.00** | 0.64 | **1.00** | **1.00** | **1.00** |
| Playground | Abs Rel↓ | Dyn | 0.35 | 0.52 | **0.08** | 0.16 | 0.16 |
| | | All | 0.30 | 7.67E+03 | **0.07** | 0.15 | 0.08 |
| | $\delta < 1.25$ ↑ | Dyn | 0.36 | 0.49 | **0.93** | 0.64 | 0.89 |
| | | All | 0.44 | 0.59 | **0.96** | 0.78 | 0.94 |
| | $\delta < 1.25^2$ ↑ | Dyn | 0.61 | 0.72 | **0.99** | 0.98 | 0.94 |
| | | All | 0.71 | 0.78 | **0.98** | 0.91 | 0.97 |
| | $\delta < 1.25^3$ ↑ | Dyn | 0.67 | 0.83 | **1.00** | 0.98 | 0.94 |
| | | All | 0.82 | 0.87 | **0.99** | 0.98 | 0.98 |
| Skating | Abs Rel↓ | Dyn | 0.16 | 0.24 | 0.15 | 0.12 | **0.10** |
| | | All | 0.10 | 1.09 | **0.02** | 0.05 | 0.04 |
| | $\delta < 1.25$ ↑ | Dyn | 0.89 | 0.59 | 0.76 | **0.93** | **0.93** |
| | | All | 0.92 | 0.29 | 0.99 | **1.00** | 0.99 |
| | $\delta < 1.25^2$ ↑ | Dyn | **1.00** | 0.95 | 0.97 | 0.97 | 0.97 |
| | | All | **1.00** | 0.42 | **1.00** | **1.00** | **1.00** |
| | $\delta < 1.25^3$ ↑ | Dyn | **1.00** | 0.97 | **1.00** | **1.00** | 0.97 |
| | | All | **1.00** | 0.53 | **1.00** | **1.00** | **1.00** |
| Truck | Abs Rel↓ | Dyn | 0.32 | 0.13 | **0.03** | 0.14 | 0.08 |
| | | All | 2.06E+06 | 0.22 | **0.03** | 0.14 | 0.05 |
| | $\delta < 1.25$ ↑ | Dyn | 0.26 | 0.81 | 0.99 | 0.94 | **1.00** |
| | | All | 0.50 | 0.71 | **1.00** | 0.79 | **1.00** |
| | $\delta < 1.25^2$ ↑ | Dyn | 0.99 | 0.99 | **1.00** | **1.00** | **1.00** |
| | | All | 0.88 | 0.94 | **1.00** | **1.00** | **1.00** |
| | $\delta < 1.25^3$ ↑ | Dyn | **1.00** | **1.00** | **1.00** | **1.00** | **1.00** |
| | | All | **1.00** | 0.99 | **1.00** | **1.00** | **1.00** |
| Umbrella | Abs Rel↓ | Dyn | 0.08 | 0.51 | **0.02** | 0.03 | **0.02** |
| | | All | 0.10 | 1.62E+06 | **0.02** | 0.05 | **0.02** |
| | $\delta < 1.25$ ↑ | Dyn | 0.91 | 0.87 | **1.00** | **1.00** | **1.00** |
| | | All | 0.84 | 0.68 | **1.00** | 0.99 | **1.00** |
| | $\delta < 1.25^2$ ↑ | Dyn | **1.00** | 0.91 | **1.00** | **1.00** | **1.00** |
| | | All | **1.00** | 0.71 | **1.00** | **1.00** | **1.00** |
| | $\delta < 1.25^3$ ↑ | Dyn | **1.00** | 0.92 | **1.00** | **1.00** | **1.00** |
| | | All | **1.00** | 0.72 | **1.00** | **1.00** | **1.00** |
| Mean | Abs Rel↓ | Dyn | 0.19 | 2.76E+04 | **0.05** | 0.10 | 0.06 |
| | | All | 2.58E+05 | 2.70E+05 | **0.03** | 0.08 | **0.03** |
| | $\delta < 1.25$ ↑ | Dyn | 0.69 | 0.59 | 0.95 | 0.93 | **0.97** |
| | | All | 0.75 | 0.58 | **0.99** | 0.94 | **0.99** |
| | $\delta < 1.25^2$ ↑ | Dyn | 0.95 | 0.73 | **0.99** | **0.99** | **0.99** |
| | | All | 0.95 | 0.72 | **1.00** | 0.99 | **1.00** |
| | $\delta < 1.25^3$ ↑ | Dyn | 0.96 | 0.83 | **1.00** | **1.00** | 0.99 |
| | | All | 0.98 | 0.80 | **1.00** | **1.00** | **1.00** |
| STD | Abs Rel↓ | Dyn | 0.10 | 7.81E+04 | **0.05** | **0.05** | **0.05** |
| | | All | 7.28E+05 | 5.69E+05 | **0.02** | 0.04 | **0.02** |
| | $\delta < 1.25$ ↑ | Dyn | 0.29 | 0.37 | 0.08 | 0.12 | **0.04** |
| | | All | 0.20 | 0.21 | **0.01** | 0.10 | 0.02 |
| | $\delta < 1.25^2$ ↑ | Dyn | 0.14 | 0.37 | **0.01** | **0.01** | 0.02 |
| | | All | 0.10 | 0.21 | **0.01** | 0.03 | **0.01** |
| | $\delta < 1.25^3$ ↑ | Dyn | 0.12 | 0.28 | **0.00** | 0.01 | 0.02 |
| | | All | 0.06 | 0.16 | **0.00** | 0.01 | 0.01 |

Table 11: **Depth accuracy for out-of-domain data [62].** We show a comparison to previous methods on the predicted depth for the point trajectories compared to their GT depths. We compare 2 ways of running our method. Ours (C&D): Our inference time outputs for a model that was trained on cats and dogs. Ours (C&D) FT: The outputs of our model trained on cats and dogs, after fine-tuning our losses for each specific video. As can be seen, fine-tuning improves our accuracy even more.

| | | DROID-SLAM[48] | ParticleSfM[66] | RCVD[25] | CasualSAM[64] | Ours (C) | Ours (C)+BA | Ours (C)+FT |
|---|---|---|---|---|---|---|---|---|
| | ATE(mm) | **2.87** | 5.81 | 1.7E+02 | 5.57 | 21.47 | 4.17 | 4.14 |
| Balloon1 | RPE T.(mm) | **4.58** | 6.51 | 2.5E+02 | 4.96 | 27.73 | 6.76 | 6.76 |
| | RPE R.(deg.) | **0.05** | 0.08 | 3.35 | **0.05** | 0.40 | 0.07 | 0.07 |
| | ATE(mm) | 7.81 | 13.51 | 3.5E+02 | **7.74** | 41.25 | 10.22 | 9.92 |
| Balloon2 | | | | | | | | |

| | | | | | | | | |
|---|---|---|---|---|---|---|---|---|
| | RPE T.(mm) | 12.82 | 14.16 | 3.9E+02 | **11.47** | 77.12 | 16.80 | 16.78 |
| | RPE R.(deg.) | 0.13 | 0.13 | 3.21 | **0.11** | 0.84 | 0.17 | 0.17 |
| DynamicFace | ATE(mm) | **2.80** | 9.71 | 1.1E+02 | 3.59 | 32.79 | 4.11 | 3.73 |
| | RPE T.(mm) | **1.71** | 7.93 | 2.4E+02 | 2.05 | 48.39 | 3.20 | 3.16 |
| | RPE R.(deg.) | **0.04** | 0.17 | 3.34 | 0.05 | 1.04 | 0.07 | 0.06 |
| Jumping | ATE(mm) | **7.65** | 13.31 | 2.8E+02 | 7.74 | 24.24 | 8.38 | 8.61 |
| | RPE T.(mm) | 10.25 | 11.27 | 2.8E+02 | **8.69** | 36.34 | 11.35 | 12.13 |
| | RPE R.(deg.) | **0.05** | 0.06 | 3.09 | **0.05** | 0.21 | 0.07 | 0.08 |
| Playground | ATE(mm) | 7.62 | 85.47 | 1.1E+02 | 5.45 | 27.44 | 6.47 | **5.06** |
| | RPE T.(mm) | 9.51 | 90.10 | 3.3E+02 | 7.68 | 40.28 | 8.00 | **6.42** |
| | RPE R.(deg.) | **0.10** | 0.75 | 4.69 | 0.11 | 0.40 | **0.10** | **0.10** |
| Skating | ATE(mm) | **7.21** | 19.37 | 78.24 | 7.28 | 27.57 | 9.21 | 8.88 |
| | RPE T.(mm) | **8.64** | 24.76 | 3.2E+02 | 8.65 | 45.02 | 11.19 | 11.44 |
| | RPE R.(deg.) | **0.04** | 0.15 | 3.91 | 0.05 | 0.24 | 0.07 | 0.07 |
| Truck | ATE(mm) | 22.55 | - | 1.1E+02 | 17.70 | 42.47 | 19.49 | **17.53** |
| | RPE T.(mm) | 31.68 | - | 3.6E+02 | 30.24 | 69.22 | 34.61 | **28.37** |
| | RPE R.(deg.) | 0.06 | - | 2.77 | **0.05** | 0.26 | **0.05** | **0.05** |
| Umbrella | ATE(mm) | **5.20** | 39.45 | 66.01 | 7.38 | 39.27 | 7.33 | 5.99 |
| | RPE T.(mm) | 8.11 | 12.08 | 3.7E+02 | 7.01 | 39.85 | **6.98** | 8.03 |
| | RPE R.(deg.) | 0.04 | 0.05 | 3.05 | **0.03** | 0.20 | **0.03** | **0.03** |
| Mean | ATE(mm) | 7.96 | 26.66 | 160 | **7.81** | 32.06 | 8.67 | 7.98 |
| | RPE T.(mm) | 10.91 | 23.83 | 320 | **10.09** | 47.99 | 12.36 | 11.64 |
| | RPE R.(deg.) | 0.07 | 0.20 | 3.43 | **0.06** | 0.45 | 0.08 | 0.08 |
| STD | ATE(mm) | 6.25 | 28.13 | 1.0E+02 | **4.25** | 8.11 | 4.89 | 4.50 |
| | RPE T.(mm) | 9.06 | 29.82 | 55.63 | 8.60 | 16.82 | 9.86 | **7.95** |
| | RPE R.(deg.) | **0.03** | 0.25 | 0.61 | **0.03** | 0.32 | 0.04 | 0.04 |

Table 12: **Camera poses accuracy for out-of-domain data [62].** We show a comparison to previous methods on the predicted camera poses. We compare 3 ways of running our method. Ours (C): Our inference time outputs. Ours (C)+BA: Our inference time outputs, followed by a short Bundle Adjustment for a model that was trained only on cats. Ours (C) FT: The outputs of the model that was trained only on cats after fine-tuning our losses for each specific video

