# OpenReview forum: "Fast Encoder-Based 3D from Casual Videos via Point Track Processing"
_NeurIPS.cc/2024/Conference — NeurIPS 2024 poster_

### Official Review · Reviewer_6f3c · 2024-07-10

**Soundness:** 3
**Presentation:** 4
**Contribution:** 4
**Rating:** 7
**Confidence:** 4

**Summary:**

The paper presents TRACKSTO4D, a fast, encoder-based method for reconstructing 3D structures and camera positions from casual videos with dynamic content. It processes 2D point tracks using a single feed-forward pass, leveraging inherent symmetries and low-rank approximations. TRACKSTO4D is trained unsupervised, reducing runtime by up to 95% compared to state-of-the-art methods while maintaining accuracy. It generalizes well to unseen videos of different categories, offering an efficient solution for 3D reconstruction in applications like robot navigation and autonomous driving.

**Strengths:**

Method:

1. The method considers the symmetry of tracked points and the temporal sequence of the video. By designing an equivariant layer based on transformers and positional encoding, it effectively takes advantage of these properties.
2. It also uses a low-rank movement assumption to decompose the motion of the 2D tracked points into the global camera motion and the 3D motion of objects in the scene. This converts an originally ill-posed problem into a solvable and intuitive problem.

The method incorporates hard constraints into the design, resulting in more efficient training and more accurate and constrained results. Overall, I think the method is very solid.



Results:

1. TRACKSTO4D significantly reduces the runtime for 3D reconstruction from casual videos by up to 95%, making it highly efficient and practical for real-time applications.
2. The method generalizes well to unseen videos and different semantic categories, demonstrating robustness and versatility.
3. The use of unsupervised training without the need for 3D supervision minimizes the dependency on annotated datasets, which are often expensive and time-consuming to create.

The evaluation results are fairly comprehensive.

**Weaknesses:**

1. The paper primarily focuses on runtime and accuracy compared to state-of-the-art methods, but it lacks a detailed evaluation on other important aspects like robustness to noise or handling of occlusions. There is also limited discussion on how the method scales with increasing video length, number of objects, or more complex scenes. Scalability is crucial for deploying the method in large-scale real-world environments. In particular, I'm very curious how the chosen $K$ would affect the performance since $K$ is like an indirect descriptor of how complicated the point cloud's components are in my opinion.
2. The method's performance is heavily dependent on the quality of 2D point tracks extracted from videos. Poor quality or incorrect point tracking can adversely affect the 3D reconstruction accuracy. I'm curious how robust the method is against failing 2D point tracks.
3. While the low-rank assumption helps in reducing complexity, it may not capture the full variability of highly dynamic or complex scenes, potentially limiting the method’s applicability in such scenarios.

**Questions:**

1. How robust is TRACKSTO4D to variations in the quality of 2D point tracks? For instance, how does it perform when the point tracks contain noise, are partially occluded, or have tracking errors? Does it strictly follow the 2D point tracks, or does it try to rectify them to some extent? Understanding this robustness is crucial for assessing its applicability in real-world scenarios where perfect point tracking is often unattainable.
2. Can the method be extended to handle dense point clouds or full 3D reconstructions rather than sparse point clouds? If so, what modifications would be necessary, and what impact would this have on runtime and accuracy?

**Limitations:**

1. The method assumes that the movement patterns of the dynamic 3D points can be effectively represented using a low-rank approximation. This assumption might not hold true for all types of motion, particularly those involving complex deformations or highly non-linear dynamics.
2. There is an assumption on the motion parallax, without which the method fails to generate accurate camera poses.
3. The accuracy and robustness of the method are more or less dependent on the quality of the 2D point tracks extracted from the videos. Noisy or incomplete point tracks could adversely impact the 3D reconstruction and camera pose estimation.

---

> ### Author Rebuttal · Authors · 2024-08-07
>
> We appreciate the reviewer's positive assessment of our method, particularly the recognition of our consideration of symmetry. We're pleased that the reviewer finds our approach solid and our evaluation results comprehensive. We'll now address the specific points and questions raised in the review.
>
>
>
> **Q**:  Limited discussion on how the method scales with increasing video length, number of objects, or more complex scenes
>
> **A**: Our model shows good adaptability to various scenarios. For video length, we trained on 20-50 frames and successfully tested on both 24 frames (Nvidia Dynamic Scenes Dataset) and 50 frames (Pet test set), with good performance in both cases. The encoder might be able to handle longer sequences, though we haven't tested this. Regarding scene complexity and object count, the Nvidia Dynamic Scenes Dataset includes a range of scenes, from single-object scenes like "Skating" to multi-object scenes like "Jumping," and various motion types such as "Truck" and "Umbrella". Tables 7 and 8 in the Appendix provide per-scene errors, demonstrating our method's robustness across these diverse scenarios, with Figure 4 offering visual examples. We'll include a more detailed discussion of these aspects in the revised paper.
>
>
>
> **Q**: Changing the number of basis elements
>
> **A**: We explored the impact of varying numbers of basis elements in our ablation study (Table 3, main paper), training models from scratch with different K values. Large K (30) led to increased depth error, indicating insufficient regularization, while small K (2) resulted in high pixel reprojection error, suggesting over-regularization and limited 3D representation of 2D motion. Our chosen K (12) balances depth regularization and accurate pixel representation. We found no significant differences with nearby K values (e.g., 11).
>
> **Q**:  Low-rank assumption helps in reducing complexity, it may not capture the full variability of highly dynamic or complex scenes.
>
> **A**: We found K=12 basis elements effective for our evaluation set, balancing complexity reduction and motion representation. However, we acknowledge this fixed number may not capture all possible scene dynamics. Future work could explore automatically inferring the optimal number of bases per scene. We'll add this point to our limitations and future work section.
>
>
>
>
> **Q**:  Robustness to noise or handling of occlusions.  I'm curious how robust the method is against failing 2D point tracks.
>
> **A**: We first note that the input point tracks extracted by CoTracker [18] are far from perfect, containing noise, outliers, and occlusions. To further evaluate our robustness to tracking errors, we performed three experiments by modifying CoTracker point tracks. First, to measure noise robustness, we added Gaussian noise to the tracks with different noise levels and measured overall accuracy. Second, to assess outlier robustness, we replaced a fraction of point tracks with uniformly sampled x,y points. Lastly, we repeated the second experiment but marked outlier points as occluded by setting o=0 (defined in L97 in the main paper). The results, presented in the attached PDF (Table 1), demonstrate that our method tolerates significant tracking errors and occlusions.
>
>
>
> **Q**: Can the method be extended to handle dense point clouds or full 3D reconstructions rather than sparse point clouds? If so, what modifications would be necessary, and what impact would this have on runtime and accuracy?
>
> **A**: Modeling dense point clouds would be an interesting direction for future work. Currently, our network can handle up to about 1000 point tracks in 50 frames in one inference step when running on an NVIDIA RTX A6000 GPU with 48GB of memory. A possible extension to handle denser point clouds could involve querying point tracks iteratively while maintaining a shared state, but this approach remains to be explored. We will include this discussion in the revised version of our paper.

---

> ### Comment · Reviewer_6f3c · 2024-08-08
> **Keep my original recommendation**
>
> Thank the authors for providing more insights into their work. I find this paper solid enough to lay a foundation for future long-sequence and dense point cloud tracking system. I would like to keep my original recommendation.

---

> > ### Author Response · Authors · 2024-08-14
> >
> > We thank the reviewer for the response and the positive feedback.
> >
> > We agree with the reviewer's comment that our method opens the door to exciting future directions, particularly for the fast handling of longer videos with dense point cloud outputs. We will include this in the conclusion section of the revised version.

---

### Official Review · Reviewer_aqo3 · 2024-07-11

**Soundness:** 3
**Presentation:** 2
**Contribution:** 2
**Rating:** 5
**Confidence:** 4

**Summary:**

The paper introduces a method for fast 3D reconstruction of dynamic structures (or 4D reconstruction) from monocular video. The model is a transformer architecture that takes a set of 2D point tracks as input, and lifts them to 3D. It is learned with re-projection losses on 2d tracks without 3D ground-truth. The method is trained on COP3D (containing cat/dog videos) and evaluated on two test datasets: (1) 21 self-collected videos of dogs and cats and (2) NVIDIA dynamic scene dataset, using depth metrics and camera tracking metrics. It is shown to perform on par with an optimization-based (CasualSAM) but much faster.

**Strengths:**

- The method tries to tackle a challenging and significant problem, and presents a viable approach that works decently well and very fast.
- It does not need 3D ground-truth, or depth supervision, and shows good generalization ability to novel scenarios.
- The authors also collected a new test set that has more accurate depth ground-truth than COP3D.

**Weaknesses:**

- The presentation can be improved. Specifically, I had difficulty understanding the points made in Sec 2.1.
  - The permutation invariance of point sets has been discussed in early literature (e.g., pointnet), and it might be better to connect to that instead of starting from symmetry analysis.
  - I also don't really understand what is the relation of "linear equivariance" and cyclic group, and how that relates to the proposed architecture. Is there any cyclic operation that moves the end of a sequence to the beginning?
  - It seemed to me the final architecture is a standard transformer with some modifications. It would be good to highlight which part enforces translation equivariance.

- Since the method does not require 3D ground-truth. In theory, it can be trained using any video. However, the model is only trained on cats and dogs in practice. I'd like to encourage the authors to train it on larger scale dataset and observe whether there is any performance gain on the held-out set.
- It uses some hand-crafted shape and motion prior (low-rank). I'd be interested in seeing an analysis of how changing the flexibility of the model (by changing the number of basis) affects the results.
- The evaluation can be further enhanced by reporting 3D end-point-errors, since depth only evaluates one aspect of 4D reconstruction. Tapvid3d [A] or synthetic datasets like Kubrics, PointOdyssey could provide such ground-truth data.

[A] Koppula, Skanda, et al. "TAPVid-3D: A Benchmark for Tracking Any Point in 3D." arXiv preprint arXiv:2407.05921 (2024).

**Questions:**

- Is there any mechanism to handle the errors in co-tracker's 2d trajectory?
- Spatialtracker [B] is another relevant work, although then train on 3D ground-truth and rely on depth inputs. It would be helpful to discuss what are the pros and cons of using unlabelled/3D data to train the model.

[B] Xiao, Yuxi, et al. "SpatialTracker: Tracking Any 2D Pixels in 3D Space." Proceedings of the IEEE/CVF Conference on Computer Vision and Pattern Recognition. 2024.

**Limitations:**

Yes

---

> ### Author Rebuttal · Authors · 2024-08-07
>
> We appreciate the reviewer's recognition of the efficiency and the good performance of our approach.
>
>
> **Q**: The permutation invariance of point sets has been discussed in early literature… and it might be better to connect…
>
> **A**:  We appreciate the reviewer's suggestion. While our architecture is based on a generalization of PointNet to sets of symmetric elements [Maron et al. 2020, ICML], which we reference multiple times in the paper, we acknowledge that PointNet [Qi et al. 2017] and DeepSets [Zaheer et al. 2017] are indeed seminal works in this area. We agree that explicitly discussing these connections would improve the clarity of our presentation. In the revised version, we will add references to PointNet and DeepSets, and briefly discuss how our approach builds upon and extends these foundational ideas in the context of our specific problem of processing point tracks from videos.
>
>
> **Q**: I also don't really understand what is the relation of "linear equivariance" and cyclic group?
>
> **A**:
> In geometric deep learning, it's customary to first analyze the symmetries of the input data and then derive suitable equivariant layers. Linear equivariant layers, such as convolutions and DeepSets [Zaheer et al., 2017], have proven effective and can be derived relatively easily. This approach is exemplified by convolutional layers for images, which are equivariant to translations, modeled as a product of cyclic groups (which formally correspond to an image on a torus).
>
> In our case, we have a product of two group actions on the input point tracks:
>
> 1. Set symmetries (like in PointNet and DeepSets)
> 2. Time-shift symmetry (represented by a cyclic group)
>
> Modeling temporal signals using cyclic symmetries is standard in the field, analogous to the image case, and forms the algebraic basis for transforms like the Discrete Fourier Transform (DFT).
>
> We derive the linear equivariant layer structure using the DSS framework [Maron et al. ICML 2020]. However, we use this only as inspiration and replace convolutions with attention mechanisms and positional encoding in our final non-linear transformer-based architecture. This approach aligns with common practices in geometric deep learning [e.g., Bevilacqua et al. ICLR 2022]. We'll clarify this in the revised version.
>
>
> **Q**: It seemed to me the final architecture is a standard transformer…
>
> **A**: While our method incorporates transformer elements, it differs significantly from a standard transformer in its structure and processing:
> Our architecture operates on a 3D tensor of shape N × P × D (time steps × points × features) and processes it using two alternating operations:
>
> 1. Temporal Attention: A transformer with temporal positional encoding operates on each point track (N × D slices).
>
> 2. Set Attention: A transformer operates across all points at each time step (P × D slices).
>
> This alternating row-column processing is inspired by the DSS linear architecture [Maron et al. 2020] but as mentioned above, replaces linear operations with more expressive transformer layers.
>
> Regarding translation equivariance, we acknowledge that our final design is not formally equivariant to cyclic translations. Instead, it provides a strong inductive bias for temporal data through the use of temporal positional encoding. Our ablation studies (Table 3, original paper) demonstrate that this design significantly outperforms the strictly linear equivariant layers.
>
>
>
> **Q**:  3D errors vs depth errors for evaluation.
>
> **A**: Our evaluation focuses on depth metrics for lifted point tracks, as current state-of-the-art baselines addressing our case primarily predict dynamic depth maps. For our model, we also assess pixel reprojection error, which evaluates tracking accuracy in the other two dimensions (see Table 3, main paper).
>
>
> **Q**: Is there any mechanism to handle the errors in co-tracker's 2d trajectory?
>
> **A**: We note that the input point tracks extracted by CoTracker [18] are not perfect and contain noise and outliers. Nevertheless, as demonstrated in our paper, our model shows significant robustness to these errors.
> Our model handles imperfect input through two features:
>
> * Static scene modeling: Ensures that only 2D motion that can be represented by a camera motion and truly static points is modeled statically.  This makes our camera estimation robust to errors while pushing the non-modelled errors to the dynamic part.
>
> * Dynamic scene modeling: Using limited basis elements for dynamic parts inherently resists outliers and extreme anomalies.
>
> To quantify this robustness, we conducted tests by adding Gaussian and uniform noise to CoTracker points. The results (see Table 1 in the attached PDF) confirm that our method tolerates significant noise levels, further validating its effectiveness in handling imperfect input data. We will include this discussion in the revised version of our paper.
>
>
> **Q**: Spatialtracker [B] is another relevant work.
>
> **A**: SpatialTracker [53] is a concurrent work discussed in our related work section. Our approaches differ significantly: unlike SpatialTracker, which relies on depth estimation inputs or 3D ground-truth supervision, our fully self-supervised method requires only 2D point tracks as input, allowing us to train on a wider range of video data and fine-tune on test cases for improved accuracy (see Tables 1,2, Ours+FT in main paper).
> We have experimented with MiDaS-inferred depth as additional input but this hasn't improved performance. This remains an area for future investigation. We will add a discussion in the revised version.
>
> **Q**:  Training with more data
>
> **A**: Please see the answer to Reviewer fqXP.
>
> **Q**: Changing the number of basis elements
>
> **A**: Please see the answer to Reviewer 6f3c.

---

> > ### Comment · Reviewer_aqo3 · 2024-08-13
> >
> > I read the rebuttal and would like to keep my score. I find this work novel as it uses low-rank motion regularization to learn 3d tracking from unlabeled videos.
> >
> > The limiting factor is that the low-rank motion prior typically makes the model biased to simple motion. At the same time, as the authors observed, increasing K leads to worse results due to lack of regularization.
> >
> > Furthermore, I still think it is worth discussing using 3D ground-truth vs unlabelled video + hand-crafted motion prior.

---

> > > ### Author Response · Authors · 2024-08-14
> > >
> > > We thank the reviewer for the response and the positive feedback.
> > >
> > > **Q**: The limiting factor is that the low-rank motion prior typically makes the model biased to simple motion. At the same time, as the authors observed, increasing K leads to worse results due to lack of regularization.
> > >
> > >
> > >
> > > **A**: We found K=12 basis elements effective for our evaluation set, balancing complexity reduction and motion representation. However, we acknowledge this fixed number may not capture all possible scene dynamics. Future work could explore automatically inferring the optimal number of bases per scene. We'll add this point to our limitations and future work section.
> > >
> > > **Q**:  I still think it is worth discussing using 3D ground-truth vs unlabelled video + hand-crafted motion prior.
> > >
> > > **A**: We will add a discussion on the pros and cons of using unlabeled data and 3D ground truth data to train the model in the revised version.

---

### Official Review · Reviewer_89V2 · 2024-07-12

**Soundness:** 3
**Presentation:** 3
**Contribution:** 3
**Rating:** 6
**Confidence:** 4

**Summary:**

This paper proposes a feed-forward network that takes a set of 2D TAP curves as input, outputs the 3D curves as well as extracts the camera rigid SE(3) poses.  This paper exploits the permutation and time shift equivariance when designing the encoding network. To train the model, the main loss is the re-projection loss (similar to BA). Experiments show in-category improvement and novel category generalization.

**Strengths:**

- The reviewer really likes the paper's style, exploiting the underlying structure of dynamic scenes, although it seems the symmetries exploited here are simple.
- The problem this paper is solving is very important and meaningful for the community. Although I do see more promising ways to solve the problem (see weakness), I do like this paper's perspective on this problem.
- The model learned on cats and dogs is somehow generalizable to new categories, verifying their effectiveness.

**Weaknesses:**

- I do like the equivariance part, but given the current advances in the community, I do find that maybe 3D-track or exploiting depth models may lead to better performance instead of learning everything from scratch.
- Solving a single scene efficiently: one important loss that supports the model learning is the reprojection error, which seems to conduct an implicit BA when training the model over many sequences, the reviewer is curious whether the same techniques can work on a single video efficiently. This is to justify the necessity of the feedforward network if the same technique can fit a single video efficiently as well.

**Questions:**

Please see weakness

**Limitations:**

discussed in last paragraph.

---

> ### Author Rebuttal · Authors · 2024-08-07
>
> We appreciate the reviewer's positive feedback on our paper's style and its significance to the community. Below we answer specific questions and suggestions.
>
>
> **Q**: Maybe 3D-track or exploiting depth models may lead to better performance instead of learning everything from scratch.
>
> **A**: We explored this suggestion by using MiDaS-inferred depth as additional input to our method and applying depth loss relative to MiDaS. Currently, we haven't seen performance improvements from these approaches. However, we agree that exploiting depth models and 3D-tracking techniques is an interesting direction that is worth further investigation in future work. This could potentially lead to enhanced performance compared to learning everything from scratch.
>
>
>
> **Q**: Solving a single scene efficiently.
>
> **A**: Yes, our model can be efficiently fine-tuned on specific scenes using our self-supervised loss terms. We demonstrated in the original paper, in Tables 1 and 2 (Ours+FT) that this approach further improves our accuracies. Training from scratch on a single scene, however, often failed to converge well after 30 minutes of training for some scenes. In contrast, feed-forward inference with our pre-trained model takes only 0.16 seconds. This significant time difference justifies our choice of a feed-forward modeling approach.

---

> > ### Comment · Reviewer_89V2 · 2024-08-07
> > **Keep my original recommendation**
> >
> > After reading the reviews and rebuttals, I think this paper makes enough contribution to the community, I keep my original recommendation.

---

> > > ### Author Response · Authors · 2024-08-14
> > >
> > > We thank the reviewer for the response and the positive feedback.

---

### Official Review · Reviewer_fqXP · 2024-07-12

**Soundness:** 3
**Presentation:** 3
**Contribution:** 3
**Rating:** 6
**Confidence:** 4

**Summary:**

This paper presents TracksTo4D, a feed-forward approach for estimating 3D structure and camera poses from 2D point tracks. Authors propose a novel architecture that directly processes 2D point tracks, takes into account symmetries present in the data, and assumes movement patterns can be represented by a low-rank approximation. Notably, this approach achieves similar performance to state-of-the-art methods with significantly reduced runtime. Despite training on videos of animals, TracksTo4D generalizes to unseen videos and unseen semantic categories at inference time.

**Strengths:**

Problem Motivation. This paper clearly describes the problem they are trying to solve and explicitly outlines the inputs and outputs of their method. Learning to predict camera poses and 3D structure from 2D point tracks is a challenging and relevant problem of interest in the community. Few methods currently solve both subproblems using feed forward approaches.

Ablations. Authors extensively ablate different losses used in training their method and highlight the impact of training on different subsets of COP3D (e.g. cat vs. dog vs. cat + dog)

Clear Writing and Informative Visuals. The paper is well written and easy to follow. The provided visuals in the supplement clearly demonstrate the effectiveness of the proposed method.

**Weaknesses:**

Limited Evaluation. Although many methods do not tackle jointly predicting 3D structure and camera poses, authors evaluate methods that support a subset of these tasks. However, these evaluations can be supplemented. MiDaS is quite old at this point, and there has been significant improvements in monocular depth estimation in recent years. Instead, one should evaluate multi-view or video depth estimation methods. Other simple baselines can be included to provide more context like Chained RAFT3D [1] and Lifted CoTracker [2].

Small Test Set Size. Authors evaluate on a subset of COP3D and the NVIDIA Dynamics dataset, both of which only have a limited number of videos. Instead, authors should consider evaluating on synthetic datasets like PointOddessey [3], which can allow authors to explore other dimensions of their method like data scaling (which is unique to this method because it is much faster than competitive methods).

[1] Teed et. al. Raft3D: Scene Flow using Rigid Motion Embeddings.
[2] Karaev et. al. CoTracker: It Is Better to Track Together.

**Questions:**

Evaluating Robustness to 2D Point Tracking Noise. Although authors use CoTracker in this work, it would be interesting to characterize how this method works when evaluated using a different point tracker at test-time.

Evaluating Robustness to Speed of Dynamic Motion. Although authors highlight that their method does not perform well given rapid motion, it would be useful to quantify this to benchmark performance for future methods. It would be useful to show off performance on automotive datasets that have ground truth depth (via LiDAR) and precise camera poses.

Impact of Data Scaling. Given the speed and generalizability of this approach, it would be interesting to explore how this method scales with more data, particularly since this method doesn't require labeled 3D data.

Defining Out-of-Distribution. Since this paper takes 2D points tracks (instead of raw images) as input, its I would not consider different datasets and semantic classes to be out-of-distribution, since the 2D point tracks don't have any notion of semantics. Instead, it would be useful to benchmark on data with different speed profiles, and provide breakdown analysis by speed buckets.

Contextualizing Runtimes. Since this method runs CoTracker before processing, run times should take this into consideration as well. Although the proposed method is quite fast, I believe the 2D point tracker can be a limiting factor on real-world performance.

Supplemental Baseline Comparisons. Although SpaTracker and DuST3R (CVPR 2024) are considered concurrent work, I think this paper can be further strengthen by comparing against such methods. Note that a lack of comparison to these approaches does not influence my ratings.

**Limitations:**

Yes, authors highlight that their approach can't handle rapid motion and is limited by the tracking noise imputed by CoTracker.

---

> ### Author Rebuttal · Authors · 2024-08-07
>
> We appreciate the reviewer's positive feedback on our paper's relevance, novelty, and clarity. We will now address your specific questions and suggestions raised by the reviewer.
>
>
> **Q**: Limited Evaluation,  MiDaS is quite old. Compare to video depth estimation methods.
>
> **A**: Thank you for your feedback. We acknowledge that MiDaS has been around for some time. However, we'd like to clarify that our comparisons were made using MiDaS 3.1, specifically the dpt_beit_large_512 version (Birkl et al. 2023). This is an improved version of MiDaS that utilizes the DPT architecture, representing a more current state of the method.
> To strengthen this point we also add new experiments with the Marigold depth estimation method (CVPR 2024). Our experiments show that our model achieves higher accuracy compared to both MiDaS 3.1 and Marigold. We've included a table with additional comparison to Marigold in the attached PDF (Table 3) for your reference. To clarify, as the reviewer asked, we performed all these comparisons using lifted CoTracker tracks. We believe these additional evaluations address the concern about limited comparisons and demonstrate our method's effectiveness.
>
>
> Regarding video depth estimation methods, our submission already includes comparisons with CasualSAM and RobustCVD. It's worth noting that while RAFT3D infers 3D scene flow from two depth images, our setup assumes RGB frames as input without depth information, making a direct comparison challenging.
>
>
> **Q**:  Small Test Set Size. Run evaluation on  PointOddessey .
>
> **A**: Following the reviewer's request, we ran an evaluation on multiple test cases from PointOdyssey and compared them with CasualSAM [59], which is the most accurate baseline. The results are presented in the attached PDF (Table 2). We observed that our method generalizes well to these cases while taking much less time than CasualSAM and maintaining high accuracy. We will make an effort to extend the evaluation on PointOdyssey to include more test cases in the final paper.
>
>
> **Q**:  Robustness to 2D Point Tracking Noise.
>
> **A**: We first note that the input point tracks extracted by CoTracker [18] are far from being perfect, and contain noise and outliers. To further evaluate our robustness to noise, we added additional Gaussian noise to CoTracker points and measured the final error. We observed that our method tolerates a significant level of noise. See Table 1 in the attached PDF.
>
>
> **Q**: Using a different point tracker at test time.
>
> **A**: We followed the reviewer's suggestion and tested point tracks extracted by TAPIR [10]. Although we observed some degradation in accuracy, the results are still good especially after finetunning, demonstrating our method's robustness to different point trackers. We added these results to the attached PDF (Table 4) and will add them to the revised paper as well.
>
>
> **Q**:  Robustness to Speed of Dynamic Motion
>
> **A**: To clarify, our limitation to the speed of dynamic motion comes from CoTracker's [18] limitation to track fast motion. In our evaluation set, we did not see such a limitation including in the Nvidia Dynamic Scenes Dataset [58] which contains a variety of motion characteristics.
>
>
> **Q**:  Automotive datasets that have ground truth depth
>
> **A**: We observed that CoTracker [18] often fails when applied to automotive scenes due to high motion blur, especially on the road. This prevents us from running large-scale evaluations on automotive datasets with the current performance of the tracking methods. We will clarify this in the revised paper.
>
>
>
> **Q**: Training with more data
>
> **A**: We observe that our method, though trained only on pet videos, already generalizes well to scenes from the Nvidia Dynamic Scenes Dataset [58], as shown in the main paper. During the rebuttal period, following the reviewer's request, we also tested our model on a variety of samples from the large-scale PointOdyssey dataset (see Table 2 in the attached PDF). Our results demonstrate that the model already performs well on this dataset without additional training. Due to the dataset's large scale, we were unable to train on it during the rebuttal period. For the revised version, we will make an effort to further train our method on PointOdyssey samples and report the effects of this larger-scale training.
>
>
>
> **Q**: Defining Out-of-Distribution data. Benchmark on data with different speed profiles
>
> **A**:The Nvidia Dynamic Scenes Dataset [58] contains several speed profiles, e.g. 'Skating', 'Truck', and 'Umbrella'. Tables 7 and 8 in the Appendix show per-scene errors and demonstrate the robustness to dynamic motion levels. For example, our depth accuracy on 'Skating' is slightly better than on 'Truck'. We will discuss this in the revised paper.
>
>
>
> **Q**: Contextualizing Runtimes. Tracking run times should be taken into consideration as well.
>
> **A**: Our method's runtime in Tables 1 and 2 already includes the point tracking times. Table 4 in the Appendix shows the separation of tracking time that is done as a preprocess by [18] (8.6 seconds), inference time of our network (0.16 seconds), and Bundle Adjustment time (0.24 seconds). We will clarify this in the paper.

---

> > ### Comment · Reviewer_fqXP · 2024-08-08
> >
> > Authors have sufficiently addressed my questions. I recommend this paper be accepted.

---

> > > ### Author Response · Authors · 2024-08-14
> > >
> > > We thank the reviewer for the response and the positive feedback.

---

### Author Rebuttal · Authors · 2024-08-07

We sincerely thank the reviewers for their thoughtful feedback and constructive suggestions. We were happy to see that all reviewers gave us positive ratings and appreciated the positive comments on our method's efficiency, performance, and generalization ability highlighted by multiple reviewers. As Reviewer aqo3 noted, our approach "presents a viable approach that works decently well and very fast" and "shows good generalization ability to novel scenarios." Reviewer 6f3c stated that “the method is very solid” and appreciated our consideration of symmetry and the comprehensiveness of our evaluation results.

As suggested by the reviewers, we conducted additional experiments whose results can be found in the PDF. We provide here a short summary:

* Tracking Error (Table 1): We experimented with varying levels of Gaussian and uniform noise to CoTracker points, demonstrating our method's tolerance to significant noise levels (in response to the questions raised by Reviewers 6f3c, fqXP, and aqo3 regarding robustness to noise).

* Synthetic data (Table 2): We evaluated test samples from the large-scale PointOdyssey dataset, showing good generalization (as suggested by Reviewer fqXP).

* Marigold depth (Table 3). We added a comparison to the Marigold depth estimation method (CVPR 2024), to address reviewer fqXP’s concern, and demonstrated that our method is more accurate in terms of depth accuracy.

* Robustness to point tracking method (Table 4):  As suggested by Reviewer fqXP, we evaluated our method with point tracks extracted by TAPIR [10] rather than the point tracks of CoTracker [18] which were used for training our method.  This demonstrates our generalization to the point tracking method.

---

### Decision · Program_Chairs · 2024-09-25

**Decision:**

Accept (poster)

**Comment:**

## Summary

TracksTo4D is a feed-forward approach for estimating 3D structure and camera poses from 2D point tracks. It uses a novel architecture that directly processes 2D point tracks, considers symmetries, and assumes movement patterns can be represented by a low-rank approximation. The approach achieves similar performance to state-of-the-art methods with significantly reduced runtime. The method generalizes well to unseen videos and unseen semantic categories at inference time. The method is trained on COP3D and evaluated on two test datasets. TRACKSTO4D is a fast, encoder-based method for reconstructing 3D structures and camera positions from casual videos with dynamic content.

## Strengths
- The paper aims to predict camera poses and 3D structure from 2D point tracks.
- The method considers the symmetry of tracked points and the temporal sequence of the video.
- It uses a low-rank movement assumption to decompose the motion of the 2D tracked points into the global camera motion and the 3D motion of objects in the scene.
- Few methods currently solve these subproblems using feed forward approaches.
- The authors extensively ablate different losses used in training their method.
- The impact of training on different subsets of COP3D is highlighted.
- The paper is well written and easy to follow
- TRACKSTO4D significantly reduces the runtime for 3D reconstruction from casual videos by up to 95%.

## Weaknesses
- The paper primarily focuses on runtime and accuracy compared to state-of-the-art methods.
- The method's performance is heavily dependent on the quality of 2D point tracks extracted from videos.
- The low-rank assumption may not capture the full variability of highly dynamic or complex scenes.
- Authors evaluate methods supporting a subset of 3D structure and camera poses.
- MiDaS is outdated and monocular depth estimation has improved.
- Multi-view or video depth estimation methods should be evaluated.
- Other baselines like Chained RAFT3D and Lifted CoTracker can provide more context.
- Authors evaluate on a subset of COP3D and the NVIDIA Dynamics dataset, which have a limited number of videos.
- Authors should consider evaluating on synthetic datasets like PointOddessey to explore other dimensions of their method.
## Overall
Based on the discussion, the paper should be accepted once all the points raised by the reviewers are addressed in the final version of the paper